

**Ice buttressing-controlled rock slope failure on a cirque headwall,**
**English Lake District**
Paul A. Carling[a, b,] John D. Jansen[c], Teng Su[d, e], Jane Lund Andersen[f], Mads Faurschou
Knudsen[f]
[a] Geography and Environmental Science, University of Southampton, Southampton, SO17
1BJ, UK.
[b] Lancaster Environment Centre, Lancaster University, Bailrigg, Lancaster, LA1 4YW, UK.
[c] GFÚ Institute of Geophysics, Czech Academy of Sciences, Prague, Czechia.
[d] University of Chinese Academy of Sciences, Beijing 100049, China.
[e] Laboratory of Water Cycle and Related Land Surface Processes, Institute of Geographical
Sciences and Natural Resources Research, Chinese Academy of Sciences, Beijing, 100101,
China.
[f] Department of Geoscience, Aarhus University, Aarhus, Denmark.
Corresponding author: Paul A. Carling (p.a.carling@soton.ac.uk)
**Key Points**
Geometry and mechanics of cirque rock slope failure defined from the local geology
Rock slope failed slowly due to ice buttressing the cirque headwall
Rock slope failure occurred during deglaciation





**Abstract**
Rock slope failures in the English Lake District have been associated with deglacial processes
after the Last Glacial Maximum, but controls and timing of failures remain poorly known. A
cirque headwall failure was investigated to determine failure mechanisms and timing. The
translated wedge of rock is thin and lies on a steep failure plane, yet the friable strata were
not disrupted by downslope movement. Fault lines and a failure surface, defining the
wedge, were used as input to a numerical model of rock wedge stability. Various failure
scenarios indicated that the slope would have failed catastrophically, if not supported by
glacial ice in the base of the cirque. The amount of ice required to buttress the slope is
insubstantial, indicating likely failure during thinning of the cirque glacier. We propose that,
as the ice thinned, the wedge was lowered slowly down the cirque headwall gradually
exposing the failure plane. A cosmogenic $^{10}$Be surface exposure age of 18.0 ± 1.2 ka from
the outer surface of the wedge indicates Late Devensian de-icing of the back wall of the
cirque, with a second exposure age from the upper portion of the failure plane yielding 12.0
± 0.8 ka. The 18.0 ± 1.2 ka date is consistent with a small buttressing ice mass being present
in the cirque at the time of regional deglaciation. The exposure age of 12.0 ± 0.8 ka
represents a minimum age, as the highly-fractured surface of the failure plane has
experienced post-failure mass-wasting. Considering the dates, it appears unlikely that the
cirque was re-occupied by a substantial ice mass during the Younger Dryas Stadial.

**Key words:**
rock slope failure, Pleistocene glacial cirque, cosmogenic exposure dating, deglaciation,
Younger Dryas, English Lake District.

**1      Introduction**
There are at least 70 known or suspected rock slope failures (RSFs) in the Lake District of NW
England that have been associated with the Late Devensian glaciation (Marine Isotope Stage
2: Wilson *et al*., 2004; Jarman and Wilson, 2015a). Such RSFs often are termed 'paraglacial'
as they "are part of, or influenced by, the transition from glacial conditions to non-glacial
conditions" (Ballantyne, 2002; McColl, 2012). However, the relationship between glaciation,
deglaciation, and the occurrence of RSFs remains far from resolved. This paper provides a
contribution to further understanding of the topic. Although a few highly modified



landforms have been identified tentatively as RSFs and related to time periods before the
Last Glacial Maximum (LGM; c., 26.5 ka BP to 19 ka BP, Clark et al., 2009) (Jarman and
Wilson, 2015b), the majority of Lake District RSFs have been associated with the end of the
Dimlington Stadial (see 'Glacial Context') and the final down-wasting of the Late Devensian
ice sheet within NW England.  At that time, potential RSFs could have been fully supported
or partially supported by residual ice masses in topographic lows.  Alternatively, some RSFs
could have occurred (Wilson, 2005) following the Scottish Readvance (*c*., 19.3 – 18.2 ka;
Chiverrell *et al*., 2018) and the Younger Dryas Stadial (12.9 – 11.2 ka; Rasmussen *et al*.,
2006).  However, only a few disintegrated RSFs have been dated.  In contrast, those that
represent steep-slope deformation, or arrested slides, are of unknown age (Jarman and
Wilson, 2015b).  An arrested hillslope failure occurs when the slipped mass is not evacuated
from the source area (Jarman, 2005), but is retained on the lower slope of the footwall.  The
role of glacial ice in buttressing rock slopes, and thereby preventing failure (Whalley *et al*.,
1983; Holm *et al*., 2004; Cossart *et al*., 2008; Le Roux *et al*., 2009; Allen *et al*., 2010; Hilger *et*
*al*., 2018), is largely speculative (Ballantyne, 2002; Jarman and Wilson, 2015b; Cody *et al*.,
2018; Hartmeyer *et al*., 2020) and controversial (McColl *et al*., 2010), as are the mechanics of
slope failure in situations where ice-support progressively diminishes (McColl and Davies,
2013; Klimeŝ *et al.,* 2021; Cave and Ballantyne, 2016).  The latter two generic issues are the
primary focus of this paper.

Glacial erosion can steepen cirque headwalls to the extent that faulted and/or fractured-
rock slopes become unstable (Sass, 2005; Moore *et al*., 2009), if not ice-supported.  In
addition, the way slopes fail can provide insight to whether ice was present during the slope
failure.  If ice-buttressed failures can be dated, then RSFs provide a source of information on
the timing of the final ice retreat.  Here, an arrested (*sensu* Jarman, 2005) translational RSF is
described, dated, and the likely controls on the failure are defined and modelled.  We test
the hypothesis that *a steep, faulted, and unstable rock slope has experienced buttressing by*
*glacial ice*.  Our study area, which has not been previously identified as a RSF site, is within
Great Coum (54.3923° N, 2.6057° W), a small cirque within the southern Shap Fells to the
west of the Lune gorge (Fig. 1).  A neighbouring cirque is named Little Coum.  The Lune gorge
(south of Tebay; Fig. 2) separates the southerly extension of the Shap Fells to the west from
the Howgill Fells to the east.  The site details and glacial context are described below.






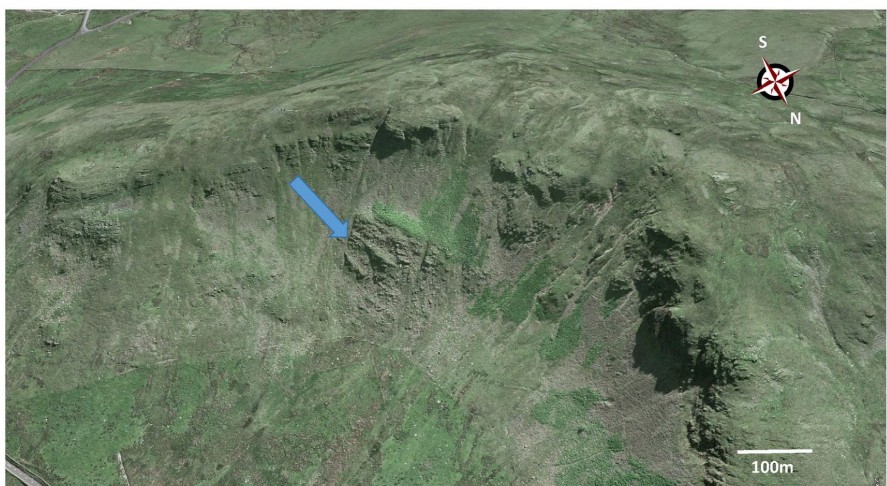


Fig. 1: Oblique aerial view, looking southwestward, into Great Coum (Google Earth image).
The RSF is arrowed. The green grassy tread of the RSF (just above the arrow) is in sunlight
below the cliffed headwall (in shadow). The breadth of the RSF is between 125m and 180m.
Little Coum is just out of view to the right. Base image © Google Earth 2014. Scale bar
applies to the middle distance.

Earth **Surface**
**Dynamics**
Discussions

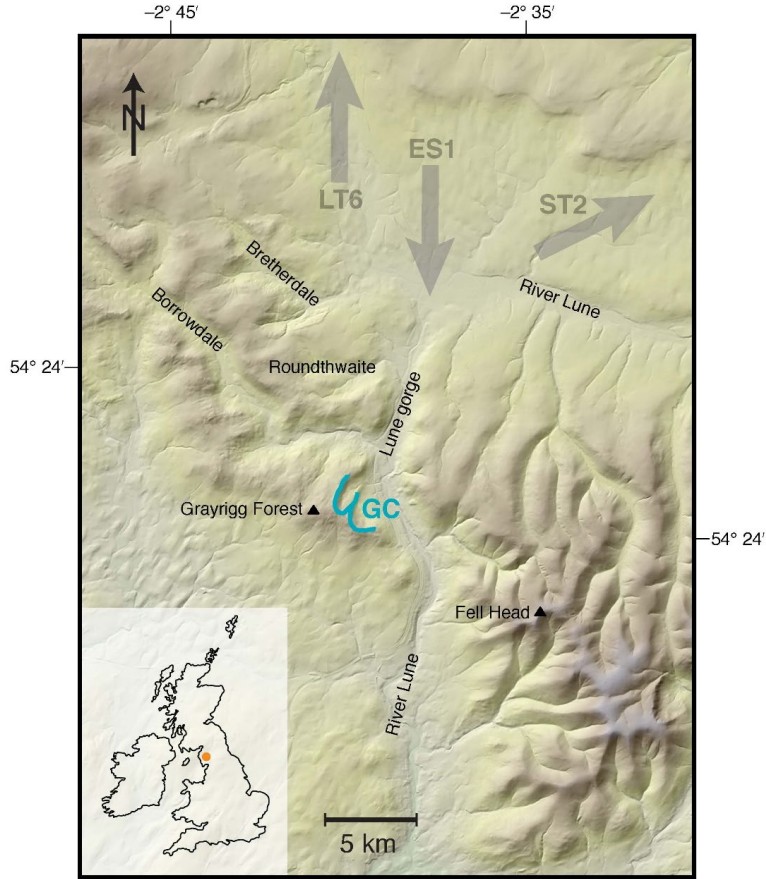



Figure 2: Regional map showing the location of Great Coum (GC) and Little Coum with
respect to generalized Dimlington (e.g., ES1) and (LT6) ice movements (after Livingstone et
al., 2010; 2012).  Locations referred to in the main text are also shown.  Inset shows the
location of the study area in the context of the British Isles. Base NEXTMap digital elevation
topography has a 5 m resolution.

**2      Glacial Context**
The last period of extensive glaciation in northern Britain occurred during the Dimlington
Stadial of the Late Devensian substage of the Pleistocene (~28-15 ka; Rose, 1985; Scourse *et*
*al*., 2009; Chiverrell and Thomas, 2010; Davies *et al*., 2019), equivalent to Stadials 3 and 2 of
the North Greenland Ice Core Project (NGRIP) chronology (Lowe *et al*., 2008) and the Marine
Isotope Stage 2 (Ehlers and Gibbard, 2013).  During the LGM, the Lune gorge and surrounds
were covered by several hundred metres of ice.  The Lune gorge is ice-sculpted, having a

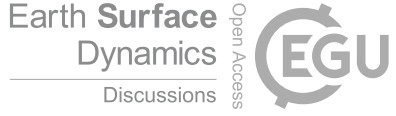

parabolic bedrock cross-section with truncated valley-side spurs along both the west and the
east margins.  Great Coum and Little Coum, on the western side of the gorge, are the only
recognized cirques in the Lune gorge.  These two Lunedale cirques should not be confused
with two cirques with the same names in Dentdale (Barr *et al.*, 2017).  The two conjoined
embayments were considered by Marr and Fearnsides (1909) to be a single cirque, but
recently have been recorded as separate cirques (Barr *et al.*, 2017; Clark *et al.*, 2018).
Devensian till banks and moraine (entrenched by the River Lune) fill much of the Lune gorge
floor and till also occurs in most tributary valleys (Aveline *et al.*, 1888; Marr and Fearnsides,
1909; BGS, undated; 2008 a and b).

**2.1    Complexity of Devensian glaciation around Great Coum**

The context in which the RSF occurred is relevant to the interpretation of the importance of
potential ice buttressing and is referred to within the Discussion.  Here we provide the
setting.  Little is known of the glacial history of the Lune gorge area (Carling *et al.*, 2023).
Nevertheless, prior findings, in the main, have been incorporated in the BRITICE maps of the
area (Stokes *et al.*, 2018).  A complex interplay occurred in the vicinity of the Lune gorge
between several upland ice dispersal centres, primarily: the Scottish, Lake District and
Howgill ice masses, during the period of maximum ice cover ~ 26–22 ka.  All three ice masses
interacted in the north whilst the latter two ice masses dominated to the south.  After the
LGM, as the ice sheets down-wasted and ice flows became increasingly valley-confined, ice
emanating from the two cirques would have flowed northwards (Carling *et al.*, 2023).

The complexity of regional ice flow was simplified by Livingstone *et al*. (2010; 2012) by using
codes to refer to different ice streams (Fig. 2) that occurred in various locations and at
differing times; the relevant codes are as follows.  In the ES1 phase, early LGM, northern ice
penetrated a short distance into the Lune gorge (Harkness, 1870; Goodchild, 1875; 1889;
Marr and Fearnside, 1909; Hollingworth, 1931; Moulson, 1966; Letzer, 1978) as far as
Carlingill and Great Coum (Fig. 2) but no further.  However, Davies *et al.* (2019)
demonstrated that, close to the LGM, (ST2 phase; *sensu* Livingstone *et al.*, 2010) and during
the LT6 phase (Chiverrell *et al.*, 2018), ice flowed northwards from the Lune gorge (Fig. 2).
On the northern flank of the Howgill Fells, any ST2/LT6 ice flow would have been to the





north and east from the Howgill ice dome (Fig. 2) such that the higher summits of the
Howgill Fells were not overrun by ice from further north (Gunson, 1966; Stone *et al.*, 2010).
Rather, the Howgill Fells hosted its own local ice dispersal centre.  Prior work failed to
determine whether northern ice entered the two Lunedale cirques.  Consistent northerly
down-wasting ice-flow was established (Hollinsworth, 1931; Rose and Letzer, 1977) from 19
ka (Davies *et al.*, 2019) with surrounding areas north and south of the Lune gorge being ice
free by ~ 19.2 to 16.6 ka (see Carling *et al.*, 2023, for a review of regional dates).  These dates
are broadly consistent with other dates for deglaciation of the central Lake District more
widely (Wilson and Lord, 2014) and are indicative of a general ~ 2–3 kyr window for the
timing of final Dimlington ice down-wasting within the Lune gorge when the back wall of the
Great Coum cirque could have become ice-free.  We return to this point in sections 3 and

153    6.2.


Given that ice may have reoccupied upland terrain in Lake District during the Younger Dryas
(Brown *et al.*, 2013; Bickerdike *et al.*, 2018), in principle, an ice mass may also have occurred
in the general vicinity of the Lunedale cirques at this time.  However, no evidence for
Younger Dryas ice in the Lune gorge has been reported.

**3        Geological Setting of the cirques**

The bedrock in the cirques comprises the marine Silurian Coniston Group (Soper, 1999;
Soper, 2006), which here consists of fine-grained, blue-grey, sandy siltstone (greywacke) in
beds from < 1 m to ~ 3 m thick.  Most of the thicker beds crop-out within the headwalls of
the cirques.  The thicker sandstone beds are more competent with fewer fractures, whilst
thinner fissile siltstone beds exhibit cleavage and are heavily fractured.  Vertical joints are
frequent with spacings of a few metres, together with evidence of small-scale bedding
deformation and small-scale faulting.  Moseley (1968; 1972) considered the considerable
complexity of the regional structure and noted folding, steep discontinuous local faulting,
joint patterns and the presence of slickenside surfaces in the southern Shap Fells.  In the
Methods and the Results, this complexity is not considered, as the detail is not pertinent to
our study.  None-the-less, reference is made to local steep faults, slickenside surfaces and
friability where these are relevant, as the rock structure in the vicinity of the RSF is critical in



assessment of slope stability (Bonilla-Sierra *et al*., 2015; Stead and Wolter, 2015).  The
apparent dips of the local beds range from 0º to 30º, SW into the headwall of Great Coum.
However, the apparent 8º plunge of the stratal sequence is towards the NW, such that the
true dip is to the WSW with a NW strike (BGS, 2008 a and b).  Infrequent, but distinctive 10–
40 mm-thick pale bands of siltstone occur (*e.g.,* Taylor *et al*., 1971, p. 26) in some of the
thicker beds, which extend discontinuously over distances of several decametres parallel to
the primary bedding.  These siltstone bands are significant in that examples (here termed
marker horizons) occur in the headwall strata which correlate with similar siltstone bands in
the strata of the RSF.

Great Coum is orientated NE, Little Coum is orientated NNE.  The orientations of the cirques
are influenced by the strike of local paired anticlines (Marr and Fearnside, 1909; BGS,
2008a), and the low-insolation aspects of both sites would have encouraged Devensian snow
and ice accumulation and preservation.  Great Coum exhibits no distinct lip (i.e., no
overdeepening), the ground falls steadily from around 237 m (height above mean sea level,
m asl) to the River Lune below (Fig. 3A).  Above 300 m the ground rises more steeply to
rocky head walls locally near 80º at 360–440 m, giving a height range of around 225 m.  Little
Coum exhibits a slight lip at around 262 m altitude.  Above 400 m the ground rises more
steeply to near 80º rocky head walls at 400–440 m, and the ridge crest at 480 m gives a
height range of around 220 m (Fig. 3B).  First to second-order minor streamlets occupy the
lower parts of Great Coum, and Little Coum is drained by the third-order stream, Burnes Gill.




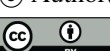

Earth **Surface**
**Dynamics**
Discussions

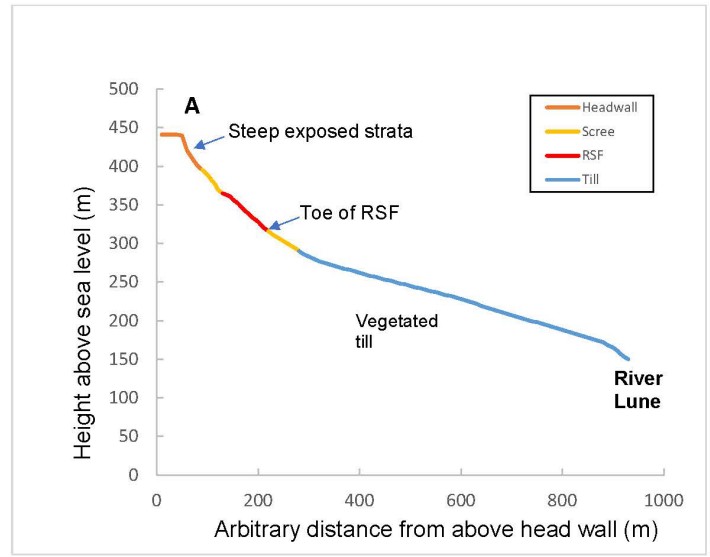

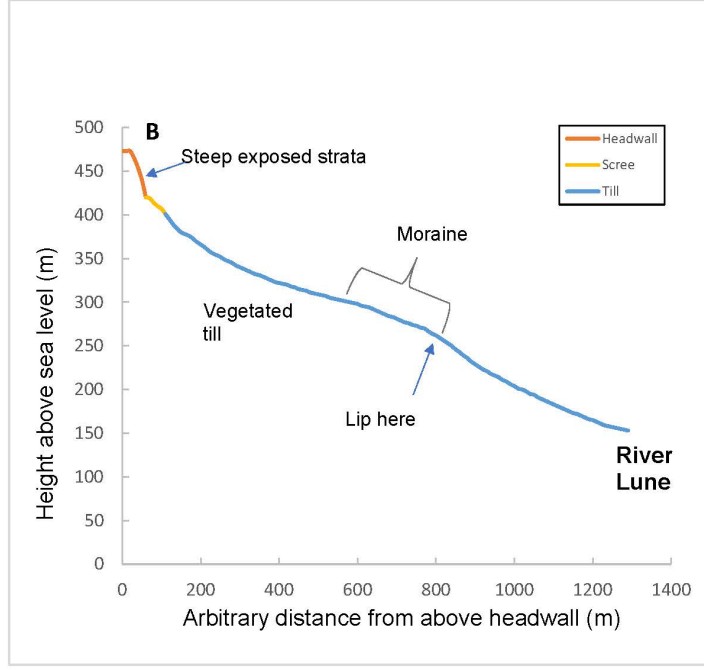


Figure 3: Long profiles along centre of cirque: (A) Great Coum; transect from 54.389978 N; 2.606625 W to 54.396675 N; 2.598278 W; (B) Little Coum; transect from 54.393789 N; 2.615992 W to 54.400117 N; 2.602208 W. Exposed bedrock is indicated in the headwall and in the RSF. Data extracted from Google Earth.




The British Geological Survey (BGS,2008b) map identifies till on the lower slopes of Great
Coum and the BGS borehole database contains the records of 24 shallow boreholes ranged
along the axis of the Lune gorge over 1.25 km immediately below Great Coum.  These
borehole logs show that the slopes just below the cirque consist of a thin soil above a 2.5 m
thickness of Devensian diamicton, overlying the Silurian Coniston Grit.  In Little Coum,
hummocky till infills the cirque below 400 m (BGS, 2008b); at lower altitudes, a flatter thin
diamicton drapes much of the basin, including a poorly defined curvilinear moraine that
terminates at the lip (Fig. 3B).  It is not possible to calculate an equilibrium line altitude (ELA)
with any certainty based on the upper limit to highest lateral limits to the curvilinear
moraine (Porter, 2001), but 300 m asl is a reasonable estimate.  Bedrock exposures along
Burnes Gill and augering during the current project indicate that this moraine is no more
than 6 m thick.  The curvilinear moraine has a distinctive, sharp, outer margin along the
rocky rounded ridge that separates the two cirques.  Within Little Coum, three faint
diamicton-covered (possibly ice-recessional) benches occur on the northern slope of the
cirque.  Thus, although Great Coum lacks any preserved indication of ice retreat, such
indicators may exist within Little Coum.

All the deposits described above are significant.  In the first instance, substantial till in the
Lune gorge below Great Coum has been related to northern ice penetrating the gorge
around the LGM (Carling, *et al*., 2023).  At that time, the whole region was covered by a thick
ice sheet (Merritt *et al*., 2019).  However, as down-wasting led to increasing topographic
control and valley glaciers predominated, there was likely to be ice flow out of the cirques
prior to the near-complete ice retreat that left the diamicton-covered benches. We envisage
that around the LGM, thick overriding ice in the vicinity of Great Coum was dictated by
regional ice gradients largely independent of the local topography (Carling *et al*., 2023).
Post-LGM, ice-discharge from the cirque initially would have remained high but any
buttressing effect on the headwall would decline as the ice thinned.

The RSF occurred in Great Coum.  The most southerly backwall section of the cirque consists
of a steep rocky headwall facing N, whilst to the west a further steep rocky headwall faces
NE; a steep grassy slope occurs between these two outcrops.  The RSF caused headwall
retreat in the vicinity of the present grassy slope, leaving the intact steep rocky sections of



the backwall to either side, but the failure also extends below the north-facing headwall (Fig.
1).  Indistinct, small RSFs also occur to the east and west, which are not considered further.
Little Coum also contains a steep rocky headwall, but with no evidence of slope failures.  The
mass of the RSF in Great Coum appears to have descended as a translational near-intact
block.  Although a near-vertical fracture occurs in the right-hand side of the slipped mass
(Fig. 4), in other respects the undisturbed strata within the block readily correlate with strata
in the headwall above.

**4        Materials and Methods**
**4.1      Mapping landscape features**
The British Geological Survey (BGS, 2008a) records several lineaments in the vicinity of
Great Coum that represent small faults or large block joints.  Google Earth satellite images
(2004, 2009, 2011 and 2014) were used to visually identify these linear landscape features
as well as others of relevance (not recorded by the BGS).  Lineaments trace topographic
discontinuities, stratigraphic offsets, vegetation differences and slickensides, and these
forms were checked in the field.  Smaller-scale linear features consist of the silt banding
marker beds, and numerous minor joints (the latter not mapped).  The various points of
interest were recorded as single point data in the field using a hand-held Garmin global
positioning system (GPS).  The strikes of bedding and the direction of faults were recorded
as compass bearings whilst the dips of bedding and faults were recorded relative to a
horizontal plane using a digital clinometer.

Single point data are precise in planview whereas linear features, between two or more
well-determined points, provide the general trend of features such as gullies and faults.
GPS coordinates also were used to map the extent of the slumped block.  Due to the
inaccuracy of hand-held GPS-derived altitudes, the planview GPS coordinates were used to
determine the altitude of each point from Google Earth, and these were taken as definitive
(error < 4%) after cross-checking with Ordnance Survey 1:50,000 maps (Harley, 1975).
Selected topographic profiles were also developed from Google Earth imagery by reading $x$,
$y$ and $z$ coordinates at 10m horizontal spacings along selected planview lines running from
the top of the headwall of each cirque, across the free face and the slope below.  Finally, a
systematic search was made within both cirques for Shap granite or limestone erratics to



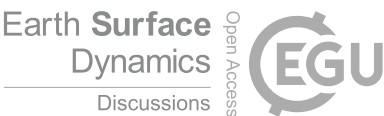

check whether northern ice had entered the cirques.  Outcrops of both these lithologies
occur 10km to the north.

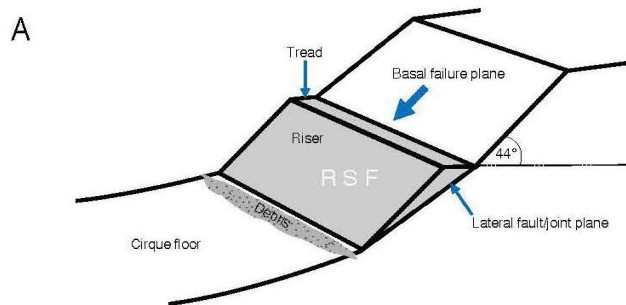

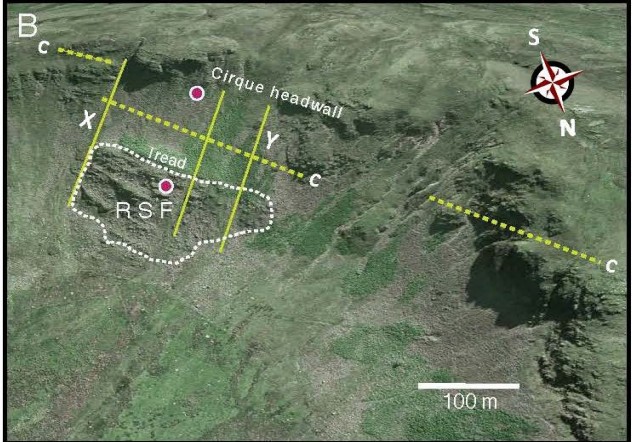


Figure 4: A) Schematic cartoon of a simple wedge failure to indicate the terminology used
within the main text. B)  Annotated view of Great Coum (compare Fig. 1). The fault-aligned
rock slope failure plane (c), above and behind the RSF, is intersected by three major steep
fault lines, the outer two of which (X,Y) define the RSF model.  The locations of samples (HW
and OSF) collected for exposure dating are shown by circled symbols.  Base image © Google
Earth 2014. Scale bar applies to the middle distance.

We refer the reader to Figure 4A for an explanation of the RSF terminology used here,
although the failure planes bounding the wedge are omitted for clarity.  The modern
headwall of the cirque locally constitutes the main exposed scarp of the failure plane behind
the translational wedge of the RSF.  The outer face of the wedge is termed the 'riser' and the
near-horizontal head of the wedge is termed the 'tread'.




### 4.2     Rock sampling for surface exposure dating

Terrestrial cosmogenic radionuclides, such as [10]Be, are produced and accumulate in minerals
within a few metres of Earth's surface due to their exposure to secondary cosmic rays and
are lost via erosion and radionuclide decay (Lal, 1991).  In our case, two free rock surfaces
are recognised (Fig. 4): the riser, being the outer surface of the RSF, and the tread, which
forms the top surface of the slipped mass.  We set out to determine when the RSF riser was
first exposed to cosmic rays as ice receded from the cirque.  For reasons of economy, we
collected one sample from the riser to compare the exposure age with the timing of
regional deglaciation. A ~15 kg intact block of bedrock (sample OSF) was collected from the
outer 10 cm-thick surface of the riser (Fig. 4B); a prominent thick undisrupted stratum close
to the top of the RSF mass. Our sampling strategy was restricted by the ease of access and
by the nature of the bedrock surfaces. The smooth bedrock surface of the riser we sampled
suggests minimal loss of rock mass due to surface fragmentation or spalling since the RSF
occurred.  A second 15 kg bedrock block (sample HW) was collected from the failure plane
of the transverse fault line just below the cirque headwall (Fig. 4B) with an aim to
determine the timing of the failure.  The sampled bedrock failure plane was observed to be
densely fractured, suggesting some loss of material from the surface since its exposure.
Surface erosion affects the abundance of cosmogenic nuclides and the estimated exposure
age; an issue we address in the Discussion (and Supplementary Materials).  Samples were
cut from *in situ* bedrock surfaces using a powered rock saw, and their altitude, bearing, tilt,
and topographic shielding were recorded.  Topographic shielding is significant for both
samples; details are given in Fig. S1 (Supplementary Materials).

Our approach entails three important assumptions about the last glaciation.  First, we
assume that the cirque headwall experienced at least 2 m of bedrock glacial erosion, which
removed the nuclide inventory produced during preceding ice-free periods; and second, that
ice burial depth at the position of the OSF sample was at least 20 m and therefore sufficient
to effectively halt nuclide production and third, that this cover persisted until failure.  These
assumptions mean that sample OSF began accumulating [10]Be only from the time that the
down-wasting ice exposed the surface to cosmic rays.  In contrast, sample HW remained



deeply shielded (> 5 m) within the cirque headwall until the RSF exposed the failure plane
to cosmic rays.

**4.3    Cosmogenic nuclide analysis**
The two bedrock samples were prepared for cosmogenic $^{10}$Be analysis at the Aarhus
University Cosmogenic Nuclide Laboratory, Aarhus, Denmark, following standard laboratory
procedures as described in Andersen *et al*. (2020). The $^{10}$Be/$^9$Be ratios were analysed at the
accelerator mass spectrometer at AARAMS, Aarhus, Denmark. A summary of the
cosmogenic nuclide analyses is given in Table 1 and further details are found within
Supplementary Materials.

**Table 1.** Summary of the cosmogenic nuclide analyses. $^{10}$Be concentrations in quartz
normalized to the "07KNSTD" standardization by Nishiizumi *et al*. (2007), and exposure ages
calculated using LSDn scaling (Lifton *et al*., 2014) and global calibration dataset (Borchers *et*
*al*., 2016) via http://hess.ess.washington.edu v3.0.2. The analytical uncertainty includes AMS
error on measured ratios incl. standard uncertainty of 1.1. %, Be carrier concentration, and
processing blank propagation (<1.2 %). The total uncertainty also includes production scaling
and calibration uncertainties. Rock density was assumed as 2.7 g cm$^3$.

| Sample ID | Latitude | Longitude | Elevation | Topo shielding correction | Sample thickness | $^{10}$Be | Uncert | $^{10}$Be age | Analytic uncert | Total uncert |
|-----------|----------|-----------|-----------|--------------------------|------------------|-----------|--------|---------------|-----------------|--------------|
| | | | (m.a.s.l.) | | (cm) | (at g$^{-1}$) | (at g$^{-1}$) | (ka) | (kyr) | (kyr) |
| HW | 54.3907 | −2.6065 | 415 | 0.74 | 1 | 57499 | 1975 | 12.0 | 0.4 | 0.8 |
| OSF | 54.3917 | −2.6064 | 348 | 0.58 | 1 | 63969 | 2017 | 18.0 | 0.6 | 1.2 |


**4.4    Rock slope failure modelling**
The RSF was modelled using *Swedge* version 6.0 (2018), a specialised rock-slope stability
software package, which can analyse a five-sided block (pentahedron) as a translational
wedge-failure—whereby a rock mass slides along a persistent basal plane of failure
bounded on each side by a fault or joint plane (Hoek and Bray, 1981; Rocscience Inc., 2018).
Either, or both, laterally bounding faults can act as additional slide planes, depending on
the geometry of the problem (Fig. 4A). In our case, two surfaces are not confined by
neighbouring bedrock: the outer surface of the RSF, the riser, and the top surface of the
slipped mass, the tread (Fig. 4). As well as varying the geometry of the failure and the



roughness of the failure planes, *Swedge* has options to consider the influence of: (i) a
tension crack at the back of the failure (not shown in Fig. 4A); (ii) water in the failure planes;
and (iii) the effect of any retaining normal stress that may counter the propensity to slide.
In engineering applications, restraining normal stress is conventionally realized using steel
rock bolts, or stone and concrete structures applied to the face of the riser, especially near
the toe.  In contrast, here the issue is whether an ice mass in the cirque can buttress a slope
that is otherwise unstable, as is explored below.  In glaciated mountain environments,
permafrost (and ice segregation) can penetrate bedrock to a depth of several metres
(Andersen *et al*., 2015). Ice-filled fissures tend to be stable at temperatures below –2°C,
which gives rise to the concept of 'ice-cemented' fractures (Ballantyne, 2018).
Consequently, the possibility that permafrost stabilized the RSF failure planes is considered
in section 5.3.

*Swedge* was implemented adopting the Mohr-Coulomb failure criterion (*e.g.,* Jaeger and
Cook, 1979) pertaining to the limit equilibrium stability of a three-dimensional rock mass
using field data (Table 2).  Further details are provided in the Supplementary Materials and
within the Results.  Stability is defined in terms of a factor of safety (*F*) where *F* > 1 indicates
a stable slope and *F* < 1, a failed slope.  *F* = 1 represents a critical state.  In general terms,
the factor of safety is defined as the ratio of the forces resisting motion to the driving forces.
Driving forces include the mass of the wedge accelerated through gravity and water
pressure; the latter applied normal to each wetted plane.  Resisting forces arise from the
shear strength of the wedge sliding planes.  Any ice load on the wedge is considered only as
a weight force contribution to the normal stress.  Thus, active support due to the load of any
glacial ice (or firn) on the riser is included in the analysis as in Equation 1; where $T_n$ is the
normal component and $T_s$ is the shear component of the force applied to the riser.  Active
support is assumed to act in such a manner as to decrease the driving force in the factor of
safety calculation:

$$F = \frac{resisting\ force + T_n\ \tan\phi}{driving\ force - T_s} \qquad (1)$$


Earth **Surface**
**Dynamics**
Discussions
EGU

Unless parameter values are known exactly, a single deterministic RSF model cannot be
resolved using Equation 1.  In view of the uncertainty, in our field case, related to the exact
relationship between fault plane alignments and dips, a variety of potential failure

**Table 2.** Parameter values for RSF as determined in the field and as explored within the three model scenarios.

|  | Riser Angle ° | Tread Angle ° | Riser length (m) | Riser Bearing ° | Width of tread (m) | Breadth of RSF (m) | Failure Plane Dip ° | Failure Plane Bearing | Failed volume (m³) | Fault X Dip orientation °N | Fault Y Dip orientation °N | Fault X bearing °N | Fault Y Bearing °N | Fault X Dip ° | Fault Y Dip ° | Tension crack |
|---|---|---|---|---|---|---|---|---|---|---|---|---|---|---|---|---|
| Field | 53 | 1 | 70 | 24 | 15 | 179 | 44 | 11 | Est: 68288 | 291 | 298 | 21 | 28 | unknown | unknown | unknown |
| Model 1 | 53 | 1 | 75 | 24 | 15 | 182 | 44 | 24 | 68333 | 201 | 208 | 21 | 28 | 80 | 72 | none |
| Model 2 | 53 | 1 | 75 | 24 | 15 | 182 | 44 | 24 | 67792 | 90 | 90 | 21 | 28 | 71 | 71 | none |
| Model 3 | 53 | 1 | 110 | 17 | 15 | 125 | 44 | 11-14 | 68739 | 111 | 62 | 21 | 28 | 90 | 62 | present |


scenarios must be considered.  To narrow the number of models, we used preliminary trials
of our field-derived parameter values as input, varying both strength and slope and
geometry parameters.  Then, consideration of a range of fault plane dips allowed us to
exclude geometrically impossible configurations and those geometries that did not
resemble the geometry of the RSF.  In this manner, we devised three model scenarios that
represent the RSF in terms of shape and mass.  More than 10,000 simulations were
performed for each scenario, varying parameter values systematically (typically ± 10%) to
isolate the most probable model for each case.  The uncertainty and probability analyses
were conducted using the dedicated approaches built into the *Swedge* platform, selecting
normal distributions to describe the possible range of parameter values; for example, ± 10°
of dips measured in the field.  Finally, the buttressing effect of any glacial ice against the
potential RSF is considered by applying an external load evenly across the area of the riser
to counter any propensity for failure.

**5.0    Results**
**5.1    The rock slope failure**
The positions of the pale silt marker beds, located in the headwall and within the RSF,
indicate the RSF has moved downslope by about 110 m (*H*) vertically and up to 192m (*L*)
horizontally.  The width of the tread is about 15 m; the breadth of the slide is between 125
and 180 m and the vertical extent of the main slipped intact mass along the outer face (the
riser) is about 70 m.  Assuming the displaced block is a triangular wedge thinning towards



the toe (Fig. 4), the volume of the intact slip is ~ 68,250 m³.  Below the main slip there is an
area of disintegrated rubble which could increase the length of the riser, potentially adding ~
3 % (~ 2300 m³) to our volume estimate (Table S1 Supplementary Materials).  The value of
$H$/L is sometimes considered a mobility ratio, whereby large values of $L$ for relatively small
vertical displacement ($H$) can indicate unimpeded rapid descent and a long runout.  Given
the volume of the RSF, values of $H/L > 0.6$, as here, indicate no excessive runout (Whittall *et*
*al*., 2017; Table S1 Supplementary Materials).

The slope of the riser of the RSF mass is currently ~ 30°, that is, is similar to the static angle
of repose. This angle may suggest slow downslope movement rather than rapid failure,
which tends to produce slope angles much less than the angle of repose.  In addition, there
was no evidence of hard-rock end-point control at the toe of the slumping block to impede
its descent although the toe has rotated outwards (Fig. 4A).  The slope of the riser today is
less than the slope of the failure plane (44°), which suggests a portion of the intact wedge
may be lying above debris derived by over-running some of the disintegrated thin toe of the
wedge (Fig. 4A).  It is significant that the stratigraphic layers within the main RSF wedge
remain intact, with no evident down-slope dilation and little deformation or fracture across
the face of the slipped mass.  The apparent plunge of the strata (8 to 10° towards the north),
*i.e.,* across the face of the RSF, indicates that the western margin of the slip may have
descended slightly further downslope than the eastern margin, as the headwall strata plunge
6° to 8° in the same direction.  The outer face (Fig. 4B) of the RSF has undergone no evident
modification.


As shown in Figure 4B, a distinct  fault (BGS, 2008b), normal to the cliff face  occurs to the
east of the RSF at location X, with undisturbed stratigraphy in the headwall either side.
Slickenside structures occur along the basal failure plane (c) that continues across the cliff to
the north-west.  The fault X is aligned with the south-eastern margin of the RSF (as seen in
Fig. 4B), whilst a further fault is evident as a distinct fissure in the RSF, with another fault to
the north-west (Y).  The easterly dip of these three faults could not be determined accurately
although they are steep, consistent with the findings of Moseley (1968; 1972) for the
Coniston group in the region (see section 5.3).  The basal failure plane defined the back of





the RSF, whilst the lateral limits to the RSF model were defined by the two marginal fault
lines (X,Y).

**5.2      Estimation of original angle of the outer slope of the rock surface before failure**
To apply the *Swedge* model it is necessary to know the angle of the outer slope of the rock
face before failure.  From the geometry of the residual RSF mass, with respect to the
observed failure plane (Fig. 4B), the RSF can be considered as a translational, plane failure of
a pentahedron wedge.  Taking a side view, the geometry is triangular (Fig. 4B), so it is
possible to calculate the minimum slope of the outer rock face prior to slope failure by
repositioning the failed block further up the failure plane.  The angle of the failure plane is
taken as equal to that of the minimum angle of the slickenside surfaces, 44°, with a bearing
of between 6 and 11°.  The riser (outer face) of the RSF is 70 m in length and the tread width
is 15 m; both lengths could have been slightly larger before fracturing occurred along the
basal failure plane and at the toe of the RSF (Fig. 4A).  Given the small degree of uncertainty
with regard to the configuration of the slope before failure, the length of the failure plane
(necessarily longer than the riser of the RSF) was varied systematically at the same time as
varying the length of the riser between the measured length of 70 m and 90 m; the latter
value includes the small area of disintegrated toe (Fig. 4A).  The tread width also is varied
between the measured breadth of 15 m and a 'limit' of 20 m to allow for potential
disintegration along the failure plane at the back of the tread.  Repositioning the RSF upslope
in this manner, the slope of the outer face could have been no lower than 53° and if the
angle of the failure plane is increased beyond ~ 54°, the resulting lengths of the failure plane
and outer face become incompatible with field observations.

**5.3      The Swedge model of the rock slope failure without ice buttressing**
Initial application of the Swedge Model used the field data shown in Table 2.  We did not
model the stability of the wedge in its present position because the basal friction properties
are unknown; whether the toe of the RSF rests on rubble derived from the failure plane, or a
bedrock surface cannot be determined.  Given that the present angle of the riser is 30° and
the basal failure plane is at an angle of 44° it is assumed that the wedge is now stable (F>>1).



The slope of the riser utilized is that applicable to the rock mass before failure, as
determined in the preceding section.  The width of the tread and the lateral extent (breadth)
of the failed mass are determined from the field data.  The summit of the cirque is fairly flat
so an outward slope of 1º below horizontal is used for the tread; the model is not sensitive
to this parameter.  The angle of the failure plane is the minimum value for the slickensides to
the south-east of the RSF (which were not disturbed by the slope failure).  The model allows
for defining the additional effective roughness angle ($r$) on the failure planes by applying a
'waviness' parameter ($w$) that was determined from the range of recorded slickenside
values, following Miller (1988).  Other parameters were defined from the field data.  It was
noted above that the dip of the two lateral delimiting faults could not be determined in the
field.  However, as local faults tend to be steep (Moseley, 1968; 1972) the model was
implemented with the values shown in Table 2 and then varied systematically as reported
below.  Given the geometry of the problem only three modelling scenarios are necessary to
explore the uncertainty in a controlled setting:

*Model 1: the RSF slides over the basal plane and against Fault Y.*  The model aligns

the compass orientation of the basal failure plane with the orientation of the riser

outer face, which assumes a simple downslope slide.  The orientation of Faults X and

Y with respect to north are as determined from field data.  The X and Y fault dips are

steep and both dip to the west.  Dips and riser length were varied slightly to optimize

the failed volume of the RSF to match the field estimate.  In this manner, the model is

not consistent with the eastern side of the slip having progressed less far down the

failure surface than the western side.  Factor of Safety: 0.83.

*Model 2: the RSF slides along the basal plane and against Fault X.* The model aligns

the bearing of the failure plane with the bearing of the slickensides to the east of the

RSF, as these define the bearing of the basal failure plane that differs from the

bearing of the riser face by 13º.  The bearings of Faults X and Y are as determined

from field data.  The fault dips are steep and both dip to the east.  Dips and riser

length were varied slightly to optimize the failed volume of the RSF to match the field

estimate.  In this manner the model is consistent with the eastern side of the slip



having progressed less far down the failure surface than the western side.  Factor of
Safety: 0.86.

*Model 3: explores the addition of a tension crack to the back of the RSF*.  It is not
known if a tension crack developed in the actual rock mass before failure, and the
properties of the tension crack are determined by the other model attribute values.
Including a tension crack, the western side of the RSF extends further down slope
than the eastern side, with the lower edge of the model block having a plunge of ~
10°, equal to the plunge of the RSF strata in the field.  The bearing of the basal failure
plane is varied between 6° and 14°.  Fault dips are steep, 90° and 62° to the east.
Dips and riser length were varied slightly to optimize the failed volume of the RSF to
match the field estimate.  Given this scenario the RSF slides over the basal plane and
against Fault Y.  Factor of Safety: 0.52 to 0.83 depending on basal plane bearing.

Given that there is unavoidable parameter uncertainty, none of the above models is an exact
representation of the RSF, although Model 3 is the closest match (Fig. 5).  Yet, it is evident
that preserving the dip of the basal plane and solving to retain the mass of the failure, any
reasonable combination of data leads to a model of the failed block that resembles that seen
in nature and, in each case, the Factor of Safety is less than unity.  A sensitivity analysis
showed that, for reasonable ranges of parameter values (typically ± 10°; outwith those listed
in Table 2), usually the geometry of the potential failure did not match that observed and so
could be dismissed. Specifically, in the 10,000 simulations of each model, model parameters
could be varied (*e.g.,* by ± 5° in the case of angles), retaining a probability of slope failure of
96 %.  In most cases the factor of safety was between 0.74 and 0.94.  In a very few cases of
parameter combinations (4 %), a marginal factor of safety of between 1.07 and 1.22 is
achieved.  In the latter cases, wetting between 20 and 30 % of the fault planes surface areas,
due to percolation of meltwater, caused the slope to fail.





Earth **Surface**
**Dynamics**
Discussions

EGU

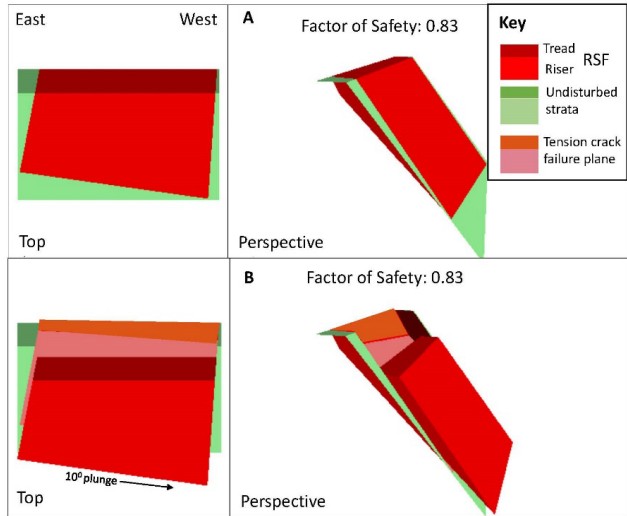


Figure 5: Illustration of Swedge 6.0 Model 3: (A) before failure, and (B) during failure. The
basal failure plane orientation is 14º such that the base of the RSF is plunging 10º to the
north (right).

As a final consideration it should be noted that in deglaciating mountain regions, RSFs have
been related to permafrost degradation and consequent destabilization of ice-filled fractures
within the rock mass (Gruber *et al.,* 2004; Gruber and Haeberli, 2007). In the RSF failure
model described above, freezing of the failure planes can be considered by simply increasing
the friction factors, which can result in the block remaining intact despite the absence of
glacial ice buttressing. However, we expect that frozen failure planes did not persist long
after glacial down-wasting. Hydrostatic pressure in the failure planes would have high, and
percolation more generally lubricates failure planes (Hasler *et al.,* 2011). In addition,
permafrost support for the RSF does not explain the intact stratification of the RSF, as
permafrost degradation would have resulted in a rapid RSF. Consequently, permafrost was
not considered in any quantitative sense.

**5.4     The Swedge model of the rock slope failure with ice buttressing**
For the range of simulations reported in the previous section, F < 1 in all the 94 % of
physically plausible cases and wetting failure planes resulted in a 100 % failure in all 30,000
cases. Hence, the role of ice buttressing of the riser must be considered, as this is the most
likely explanation for slope stabilization. There is no information on the dynamic behaviour



of ice within the cirque. Consequently, selecting Model 3 above, three contrasting scenarios
can be envisaged that might stabilize the slope: (a) ice can be a static load variably
distributed around the centroid (Fig. 6A) of the riser; (b) ice can be dynamic, moving towards
the riser such that the stress is variably distributed around the centroid of the riser (Fig. 6B);
(c) ice can be dynamic, moving away from the riser such that a bergschrund opens between
the ice and the slope and the stress is distributed below the centroid of the riser. Broadly
consistent results also are found considering Models 1 and 2 (not reported herein).

Firstly, considering scenario (a), the weight of an ice load is calculated, and the stress is
applied evenly across the area of the riser normal (*i.e.,* 90°) to the slope until it is stabilized
(for which condition: F = 1.0065; Fig. 6A). Subsequently, considering scenario (ii), the
analysis is repeated to ascertain the optimal direction to apply force that minimizes the ice
load. In scenario ii, the ice load can be reduced from that in (a) if the force is directed into
the slope and slightly upwards by $13^0$ above the horizontal such that for F = 1.0485 (Fig. 6B).
In scenario (a), application of 40,659 tonnes of ice is required for a stable slope, which is
equivalent to 48,987 $m^3$, based on a debris-free low ice-density of 830 kg $m^{-3}$ (Colgan and
Arenson, 2013). In scenario (b), application of 24,325 tonnes of ice (29,307 $m^3$) is required
for a stable slope. For scenario (c), with a tension crack, the slope will remain stable as long
as the total stress applied to the slope is the same as for scenarios (a) or (b). In this study we
do not explore in detail how the ice mass and force direction might be distributed across the
riser to maintain slope stability as there are multiple permutations. Nonetheless, if the
cirque had been filled with ice to the top of the riser, around 166,000 $m^3$ of ice would be
required to fill the volume immediately adjacent to the potential RSF (Fig. 7), which is not
compatible with the small ice masses in scenarios (a) and (b) that are required to maintain
slope stability. Considering Fig. 7, it is important to recognize that, in any permutation of
potential RSF geometry (Table 2), the ice cover required to maintain slope stability is
typically less than 29 % (and possible as low as 17 %) of the volume to the top of the riser.
This result indicates that the slope would have remained stable as long as there was a
sufficiently small degree of ice buttressing due to ice in the cirque contributing a stress
normal to the face of the riser—which further implies failure occurred during final
deglaciation of the cirque. Note that, although the presence of sufficient ice on the riser
alone maintains rock mass stability, it is unlikely that this condition would pertain without





ice present immediately adjacent to the rock wedge.  So, Figure 7 shows cirque ice beyond
the unstable slope, but only conceptually.


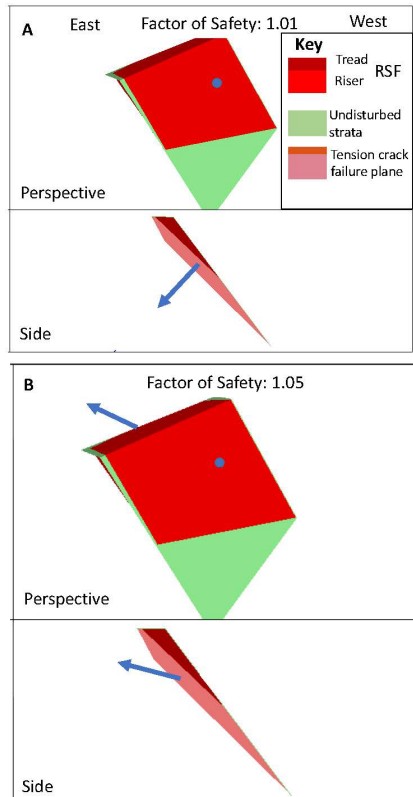


Figure 6: Illustration of the force application required to stabilize the potential RSF: (A) With
the force (point and arrow) applied 90° to the slope, the ice load required to stabilize the
slope (i.e., F = 1.0055) is 40,967 tonnes; (B) With the force (point and arrow) applied at the
optimum angle (13° above horizontal) the ice load required to stabilize the slope (i.e., F =
1.0485) is 28,253 tonnes.

Earth **Surface**
**Dynamics**
Discussions

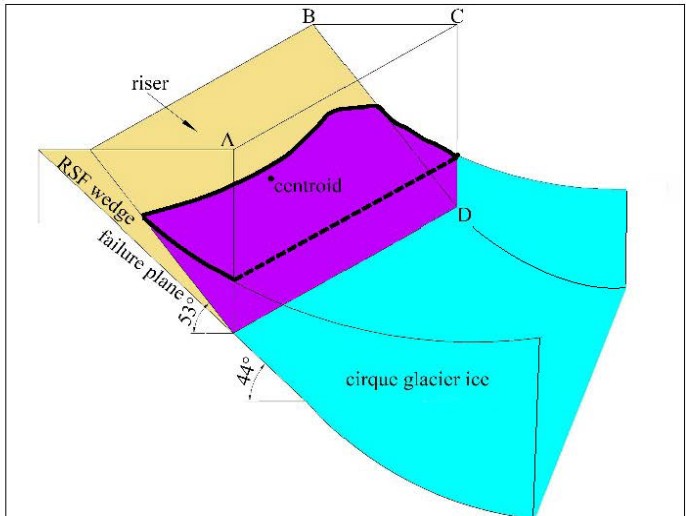


Figure 7: Cartoon depicting the concept of ice buttressing of the potential RSF. The RSF
wedge defines the unstable portion of the slope before the rock slope failure. Points A, B, C
and D define a pentahedral volume that, if filled by ice, would cover the complete face of
the riser. The pentahedral volume, with the upper ice surface outlined by a heavy black
line, shows that only a small percentage of the potential pentahedral ice volume is required
to be ice-filled to provide buttressing sufficient to prevent slope failure. Percentages were
obtained from the ice volumes required to buttress the slope. Additional ice might be
present in the cirque outside of the defined volume, but this ice does not contribute to the
stabilizing load directly applied to the riser.

596

### 5.5    Exposure ages from the rock slope failure

In both cirques, tills are composed of local lithologies exclusively, and a search for northern-

derived erratics confirmed their absence. The absence of erratic lithologies indicates that

the cirques were probably eroded by locally generated ice masses after the LGM. At the

time of the LGM, it is thought that the locations of the cirques were overridden by an ice

sheet from the north moving into the northern end of the Lune gorge (Carling *et al*., 2023).

Under such thick ice conditions, the back wall of the cirque would have been stable as the

volume of ice was much greater than that required for slope stability—shown by either ice-

loading Model 3 scenario a or b. During active cirque erosion, after the LGM, the ice

volume in the cirque would decrease such that ice-loading also decreased such that the RSF

slowly descended as the loading fell below a critical F-value to sustain the slope. It is this

lowering of the RSF that we have attempted to date with cosmogenic nuclides.




The surface exposure age of 18.0 ± 1.2 ka (sample OSF) postdates the timing of maximum ice
cover and is consistent with the timing of deglaciation within the broader region (Carling *et*
*al*., 2023) as is considered in the Discussion.  As was noted in section 4.2, the outer face of
the RSF (the riser) constitutes a smooth surface of intact, undeformed strata, so
concordance of the surface exposure age and regional dates is to be expected.  Exposure of
the RSF riser (sample OSF) predates significantly the exposure age of 12.0 ± 0.8 ka (sample
HW) calculated for the RSF basal plane, suggesting a relationship between debuttressing of
the riser face and the gradual downward slip of the RSF.  The younger age for sample HW is
expected, due to the basal failure plane being progressively exposed after the upper portion
of the RSF (where sample OSF occurs) was clear of ice cover and the RSF began to move
downslope.  Also of significance is the fact than the basal failure plane was disrupted by the
failure and is friable, as was noted in section 4.2.  The loss of only one or two small blocks
from the location sampled at any time after failure should result in an age younger than that
of the outer face of the RSF (see Supplementary material). Results of the cosmogenic nuclide
analyses are summarised in Table 1.


**6.0    Discussion**
**6.1 Modelling the RSF dynamics**
The *Swedge* model was applied to the RSF assuming the original slope of the rock face was
53°, with a slide plane angle of 44° and no ice buttressing.  The steeper slickenside surfaces
observed in the field directly above the RSF could indicate a steeper failure plane than that
used in the model, but these values were not used as they may represent strata disturbed by
the RSF.  In any case, an increase in the failure plane angle, or the initial angle of the rock
face, both increase the propensity for failure.  The waviness number calculated from field
data and applied in the model is low, which increases the propensity for failure.  Preliminary
trials showed that to stabilize unstable model slopes would require the use of unrealistically
large waviness numbers (Miller, 1988) and so the waviness number was not varied in
sensitivity analyses.  Thus, our results obtained with the *Swedge* model are conservative but
show that the rock face was consistently unstable before failure.  The sensitivity analyses
accounted for parameter uncertainty and demonstrated that, in most cases, failure would
have occurred due to gravity alone.  In those few cases where the slope was modelled as





marginally stable, moderate water lubrication of the failure surfaces (typically 30% of
surfaces) induced slope failure, but the addition of a modest amount of buttressing ice
ensured the slope remained stable.  As there is no obstacle at the toe of the RSF to impede
descent, it is reasonable to assume that the slip occurred slowly as the ice decayed.  The
need for buttressing of the slope to prevent rapid failure indicates that ice support was
important (Hilger *et al.*, 2018).  Thus, our hypothesis '*a steep, faulted, and unstable rock*
*slope has experienced buttressing by glacial ice*' as proposed in the Introduction is
corroborated here.

As the amount of Model 3 scenario (a) ice (static load normal to the face) in the cirque
decreases, the level of the ice against the riser will fall towards the toe.  Thus, the focal point
of the force applied to the slope by the ice cover migrates down the riser.  As long as the
stabilizing load and the direction of the applied force remain sufficient as ice retreats, the
detached block will remain stable.  However, the load within the cirque is unlikely to be
maintained as the ice elevation falls.  The applied force also is variable through time and
across the riser as ice primarily deforms by internal flow (Hutter, 1983) such that, if any
additional pressure were exerted by residual ice adjacent within the Lune gorge, then the ice
mass within the cirque would respond accordingly.  In particular, the uniaxial compressive
strength of ice is low and decreases as ice temperature increases, as will be the case during
deglaciation.  Although in our model we do not consider the shear stresses associated with
the ice in a quantitative sense, brittle fracture of the thin, buttressing ice mass might
ultimately occur owing to the constant pressure associated with the mass of the RSF (Bovis,
1982; McColl and Davies, 2013).  The presence of a tension crack will redistribute ice load
and induce ice segregation (frost-cracking) in the rock (Sanders *et al*., 2012) close to the toe
of the rock mass, further reducing the competency.  So as the factor of safety falls to close to
F = 1, the detached block will slowly move downwards.  In the final stages of deglaciation,
low-density firn (~400–830 kg m$^{-3}$) will replace glacier ice (~830–917 kg m$^{-3}$) offering less
support to the RSF.

The RSF failure probably was controlled by distinct intersecting small-scale faults, as has
been modelled herein.  Within the general area of Great Coum there appears to be two sets
of frequent lineaments, one trending to N to NW and the other NE, that intersect to define



bedrock blocks.  Despite this propensity, the other steep headwalls in these two cirques
show no evidence of large-scale instability, although the basal fault plane of the RSF extends
(Fig. 4) behind the more western steep buttress in Great Coum, indicating that this slope is
also potentially unstable.  One fault (BGS, 2008b) and several other lineaments occur roughly
normal to this alignment which, in conjunction, might delimit a potential wedge failure on
this western buttress.  In the specific case modelled, slope failure is highly site-specific
depending, in the main, on fault alignments.  Steepening of the cirque headwall via glacial
erosion may have altered the disposition of the rock mass load, increasing tensile stresses
along the fault planes, and promoting the RSF (Ballantyne, 2002).  In this respect, the failed
slope was pre-conditioned (*sensu* McColl and Davies, 2013) to fail.  However, the modelling
suggests that unloading likely played a role in controlling the timing of failure and the rate of
landslide displacement once initiated.  Unloading may simply allow the unsupported
preconditioned block to fail, but the stress release accompanying unloading usually is
propagated along the fault network resulting in a reduction of internal locking stresses (*i.e.*,
the waviness number; Wyrwoll, 1977; Ballantyne, 2002).  Other preparatory factors also
come into play as the ice load was removed, such as lubrication of the failure planes by
meltwater and weathering of the fault planes in general, moving the block closer to F = 1.

**6.2 Timing of the RSF**
Although there is only one terrestrial cosmogenic date for the riser of the RSF, the surface
exposure dating of 18ka is compatible with the RSF movement during final deglaciation
around 19.2 to 16.6 ka (see Carling *et al*., 2023, for a review of regional dates). We interpret
the much younger exposure age (~ 12 ka) on the fault plane as the result of postglacial
weathering and erosion. Exposure dating necessarily only yields a minimum-limiting age of
exposure, except in cases where primary structures (*e.g.*, glacial striations or slickensides)
testify to negligible surface erosion. We observed some slickensides locally preserved on the
fault plane, but some degree of surface erosion is also indicated by a scattering of talus and
a shattered basal failure plane. We provide an estimate of the magnitude of surface erosion
assuming a range of plausible erosion rates in Fig. S2, Supplementary Materials wherein the
limitations of having only two cosmogenic samples is addressed.



We note that the locally derived till and absence of northern derived erratics in the cirques
suggests that northern ES1 ice did not enter the cirques, despite the presence of abundant
(northern) Shap granite erratics in Borrowdale, Roundthwaite valley and Bretherdale just to
the north (Carling *et al.*, 2023).  Thus, buttressing of the slope by ice moving into the cirque
from the north can be ruled out.  We suggest that the two cirques probably fed valley
glaciers associated with diminishing plateau icefields after the LGM (Carling *et al.*, 2023), and
their final form evolved during deglaciation.  The Devensian termination is thought to be a
4–5 kyr period of ice decay just prior to the Last Glacial-Interglacial Transition at ~ 14.7–11.5
ka (Stone *et al.*, 2010).  During deglaciation, there was unlikely to be sufficient ice in the
adjacent Lune gorge to bolster the cirque ice mass.

Regarding slope failures in cirques, Cave and Ballantyne (2016) and Klimeŝ *et al. (*2021)
noted that the role of glacial ice support in cirque back wall stability is conditioned by the
associated time scales considered.  For example, Klimeŝ *et al. (*2021) reported high factors of
safety (> 1.95) for potential RSFs beneath glacial ice during the LGM, which is assumed to be
the case during full glacial conditions.  Ballantyne *et al*. (2014) demonstrated that, following
the LGM, the timing of several dated RSFs is not consistent with the probable timing of
glacial debuttressing, reporting ages that correspond to deglaciation and well after. In
contrast, at Great Coum, the surface exposure age of 18.0 ± 1.2 ka is consistent with regional
estimates of the timing of deglaciation (see Carling *et al*., 2023), as was noted above.
However, the apparent delay in final exposure of the fault plane, sometime before 12.0 ± 0.8
ka, indicates that a range of exposure ages might be associated with arrested RSFs; indeed,
some post-glacial dates may be associated with isostatic controls on slope failure (Ballantyne
et al., 2014).

**6.3 An ice advance during the Younger Dryas?**
An important remaining issue is whether Great Coum could have supported a glacier during
the Younger Dryas Stadial.  Although the Lake District was essentially ice-free by ~ 14.7 ka,
Younger Dryas cooling led to a subset of cirques in northern Britain refilling briefly (Evans,
1997).  Sissons (1980) argued that many central Lake District cirques were re-occupied by ice
during the Younger Dryas, and subsequent studies (reviewed by Brown *et al.*, 2011) indicate
the presence of cirque glaciers in the central Lake District.  However, the lowest Lake District





cirque floors are around 320 m asl (Temple, 1965), whereas the basal lip of Little Coum lies
at 262 m asl.  In this context, Manley (1961) argued that cirques in the Howgill Fells lack
evidence for reoccupation during the Younger Dryas because they are too low.  Norris and
Evans (2017) suggested the ELA in the western Pennines was 580 m asl during the Younger
Dryas with the lowest estimate placing the altitude at 445 m asl (Wilson and Clark, 1995).
Similarly, in the eastern Lake District, immediately to the north-west of Great Coum, the ELA
has been estimated at 400–600 m with 400 m being regarded as distinctly marginal (Wilson
and Clark, 1998).  Glacial ice only descended to altitudes below 400 m asl where small outlet
glaciers were fed from plateau icefields (McDougall, 2013), the extents of which remain
controversial (Bickerdike *et al.*, 2018).  In this respect, Harvey (1997) noted that there was
no evidence of ice readvance in the west facing Carlingill, neighbouring Great Coum.

As the top of the headwall of Great Coum is at 468 m asl, with no extensive plateau above, it
seems unlikely that snow supply was sufficient to maintain a Younger Dryas cirque glacier.
Others have also noted that Howgill cirques are too low to support Younger Dryas ice but
have suggested that the 'fresh' appearance of moraines in some Howgill and western
Pennine cirques indicate that Younger Dryas ice was maintained locally by extensive snow-
blow (Gunson, 1966; Gunson and Mitchell, 1991; Mitchell, 1996). If correct, this would
reduce the ELA locally to as low as 311 m asl (Mitchell, 1996).  Mitchell's estimate of ELA is
similar to the best estimate for Little Coum (300 m asl), and it is noted by several authorities
(Manley, 1961; Temple, 1965; Mitchell, 1996) that the dominant wind direction during the
Younger Dryas was from the W and SW, associated with cyclonic disturbances.
Nevertheless, we are not convinced by this argument.  The extensive SW-facing slopes of
Grayrigg Forest and Grayrigg Pike are below the Younger Dryas ELA, so it is unlikely that
sufficient blown-snow could have been supplied to support glacial ice within the Great and
Little Coums.  Our exposure age of 18.0 ± 1.2 ka (sample OSF) denoting ice-free conditions
on the outer face of the RSF suggests the cessation of glacial erosion at Great Coum.

**7.0    Conclusions**
We have demonstrated that a RSF in the headwall of a cirque in the Lune gorge occurred as
a slow downslope movement of an intact rock mass due to the presence of a supporting



glacial ice mass buttressing the failed slope.  The estimated RSF timing corresponds with
regional deglaciation occurring by at least 18.0 ± 1.2 ka.

Although the case study reported herein supports the role of ice buttressing as a process
which may explain arrested RSFs, the vagaries of rock structure from one location to
another, coupled with the spatially variable role of isostatic uplift and local meltwater
climate (Cave and Ballantyne, 2016) provide strong site-specific controls on the nature and
timing of RSFs.  Further modelling of RSFs should elucidate the range of conditions
associated with incipient failure whilst additional exposure ages for rock surfaces should
assist in constraining the timing during which processes such as glacial debuttressing
applied.

**Code availability**
*Swedge 6.0* is available from Rocscience Inc., Toronto ([www.rocscience.com](www.rocscience.com)) for purchase or
as a licenced educational package upon application.
**Supplement Link**
*Note to reviewer: A supplement accompanies this manuscript*
**Author Contribution**
PAC devised the project and conducted the fieldwork and the *Swedge 6.0* simulations. TS
assisted in fieldwork. PAC and JDJ wrote the manuscript.  JLA and MFK conducted the
cosmogenic nuclide analysis. All authors contributed to the final presentation.
**Competing interests**
The authors declare that they have no conflict of interest.
**Acknowledgements**
Teng Su was supported by the State Scholarship Fund of the China Scholarship Council.
Rocscience Inc., Toronto is thanked for supplying *Swedge 6.0* as an educational package.
Wishart Mitchell kindly provided a copy of the Gunson (1966) thesis.  Mike Cavanagh and
the Horned Beef Company are thanked for access permissions to collect rock samples in the
cirque.  Sam McColl is thanked for commentary on an early version of the manuscript which
contributed to the final presentation.
**Data Availability Statement**



The data required as input to *Swedge* version 6.0 (2018) are listed in Table 2.  Use of *Swedge*
version 6.0 was licensed under an educational agreement with Rocscience Ltd., 2018:
www.rocscience.com.  The [10]Be concentrations and underlying AMS data associated with the
[10]Be exposure ages are published on GitHub
https://github.com/CosmoAarhus/LakeDistrict_CosmoData.

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
