# Peer review of "Ice buttressing-controlled rock slope failure on a cirque headwall, English Lake District"

_Earth Surface Dynamics, 2023_

## Referee Comment (RC1)

**Carling et al**
**Ice buttressing-controlled rock slope failure on a cirque headwall,**
**English Lake District**

**Earth Surface Dynamics 2023**

Reviewer comments

**David Jarman**
mountain landform research
Scotland

**SUMMARY**

**Recommendation** - accept with minor changes, subject to attention to the figures
(with scope for more substantial revision/improvement at editorial/authorial discretion)

This paper addresses an apparently trivial landform feature, a very small and unimpressive RSF, which (most unusually, in the British mountains and generally) happens to be discernible from a major road (M6).  It analyses the slope stresses acting on such a slipped mass by an engineering geology technique Swedge, with and without support from glacier ice.  It dates the outer face and the scar above, and from their significantly older and younger ages, it states that this corroborates a contention that the slipped mass could only have been emplaced thus with the support of a waning valley glacier.

The paper is in fact of great interest, and a novel contribution to geomorphology, for four main reasons:
1. case studies of individual RSFs in the British mountains are uncommon, and those analysing their engineering geology are vanishingly rare (and essentially confined to a couple of PhD theses on Scottish Highlands sites, now decades old).
2. few RSFs in the Lake District (two) and Highlands (a score or so) have been dated - all of them fully disintegrated rock avalanches, whereas this is an arrested translational slide.  Even globally, quasi-intact rockslides have seldom been dated due to obvious sampling difficulties, with the emphasis on 'antiscarp' trench faces rather than outward-facing slopes as here.
3. glacier support for RSF emplacement is not an entirely original notion, but to the writer's incomplete knowledge this may be a pioneering study of a possible demonstration example, in Britain at least.
4. morphologically, RSFs in cirques form a small minority, with some of those on outer flanks rather than headwalls;  and the distance travelled by the putative slipmass while remaining quasi-intact - 110 m - is remarkable, especially for such a small feature.

(the authors are perhaps too modest here, and could increase the impact of the paper by placing its originality more firmly within the British and global montane RSF literature).

Although the site is trivial in size - at 0.03 km$^2$ in extent by a standard method it is only just above the threshold of 0.01 km$^2$ for designation as RSF (see 'References') - this proves not to be a demerit, for it is in effect a scale model or field-laboratory experiment.  Although this site is unusual in some ways, it is not *sui generis*, and the findings are relevant to the wider study of RSF behaviour.  Again, these virtues could be made clearer.

While the paper is eminently publishable essentially as it stands, it relies on a number of assumptions, stated and unstated, which bear closer scrutiny.  The paper could be greatly improved by adumbrating these assumptions;  alternatively, the editors and authors may be well content with 'putting it out there' as an aunt sally to provoke and stimulate further debate and work.  If the latter course is followed, it is recommended that at least the 'corroborated hypothesis' be expressed more circumspectly[1], with due acknowledgement that it does rest on a set of convenient assumptions, as befits a controlled experiment.

Before exploring these assumptions in the COMMENTARY that follows, some practicalities can be dealt with.

REFERENCES

The lead author is now aware of a new overview paper on Lake District RSF, published as the present paper was being submitted :
Peter Wilson & David Jarman (2022) Rock slope failure in the Lake District,
NW England: an overview, Geografiska Annaler: Series A, Physical Geography, 104:3, 201-225,
DOI: 10.1080/04353676.2022.2120261

In addition, the reference to Jarman (2005) should be 2006;  this paper is largely superseded by an overview paper:
Jarman D, Harrison S (2019): Rock slope failure in the British mountains. Geomorphology 340, 202-233.

These two papers do not require any major alterations in the present paper, beyond updating the regional data, and perhaps noting that the RSF conceptual typology has been generalised since 2005/6 for much wider montane applicability.  They do contain discussions of RSF timing and causation which may be of interest.

TEXT

The submitted paper is returned with 'sticky notes' for possible clarification, and marking a few trivial edits.  They also identify passages discussed in the Commentary below.

FIGURES

It is recommended that the visual presentation of the site be substantially upgraded:
* * *
[1] here it is noted that the paper has previously ben rejected because of its reliance on two single dates, and that the Associate Editor has already advised circumspection in this regard.  It might help further if the emphasis is taken away from reliance on these dates, and the case study presented as a multi-pronged assessment, with the dates providing a measure of comfort, but far from essential to the conclusions reached.

Fig 1    the arrow is too crude and impinges on the feature - it would be more instructive to delineate the identified RSF site boundary, including 'cavity' or source area at rim, and basal extent, with a fine dotted line, and a fine arrow suggesting slipmass trajectory

Fig 2    this regional map is neither sufficiently regional nor adequately detailed at Lune gorge context scale.  Two maps are needed:

❖ regional, including
   ➢ full named extents of Shap Fells and Howgill Fells, with Lake District as far west as say High Street;
   ➢ full extent of upper Lune basin, with general ice movements as long curvilinears, not crude arrows, and famous drumlin field just to SW identified and placed in ice outflow context;
   ➢ ice movements not just coded but numbered 1-2-3 to clarify sequence (nb. present caption omits ST2; it could helpfully specify what all three codes represent)
   ➢ extent of Silurian outcrop, with adjacent lithologies;
   ➢ RSFs, with small-large sizes, thus emphasising the High Street cluster and sparsity here (easily obtained from Wilson&Jarman 2022[2]);
   ➢ glacial cirques (from Ian Evans inventory), perhaps with grades/elevations, again emphasising sparsity around here and unusually low elevation of GC/LC;
   ➢ the M6 even !!

❖ local, perhaps as broad as present but less N-S to zoom in closer in landscape format, including
   ➢ named highest summits and elevations in each hill mass (Grayrigg Pike, not Forest, being the peak closest to the site)
   ➢ elevations along Lune gorge floor and rims
   ➢ RSFs, distinguishing parafluvial cases in Howgill V-shape valleys (#9.03 A/B) from paraglacial locations as here and Cautley #9.01/02
   ➢ cirques

Fig 3    these long profiles really require a proper map of the two cirques to locate them, which could very usefully also depict the glacial moraine features etc described - this might be includable on the 'local context map' requested above.

Fig 4A  the schematic wedge already gives one actual dimension, the 44° slope - why not add the other dimensions ?
(nb. the word 'debris' in nigh invisible)

Fig 4B  we are brought very little closer to visualising the RSF than in Fig 1, and with the same GEarth image offering no change in perspective - a different angle / historical image, or a photo, would add considerably to our grasp.
* * *
[2] four additional small sites have since been identified from imagery in this area - see appendix

(nb 1. it is not clear what the three components marked "c" intend - is a mappable fault trace actually observed to be displaced down the headwall, or exposed in it at a lower level by removal of the slipmass, or is this merely suggestive ?)
(nb 2.  sample codes HW and OSF are stated in the caption and should be added to the figure) - ideally, with the actual dates (as concise ~ages) to aid our grasping that they are older above younger - the writer has to recheck this every time....

Fig 7    this useful diagram is very hard to decipher from the caption.  Do we understand there to be two pentahedral volumes:
- potential, delineated by A-D  (can it not be extended E-F for greater clarity ?)
- needed, toned purple

The term 'pentahedral volume' should be stated as 'potential pentahedral volume' at its first appearance in the caption, and might be toned lightly.
The 'needed pentahedral volume' should be stated to be purple (why does its contact with the riser need to be curvy, in a schematic ?)

RECOMMENDED ADDITIONAL FIGURES

1. detailed site map - this map of Great Coum would define the RSF extent and components properly, in a way which the thumbnail sketch in Fig 4B cannot.  It would mark and codename the two sample locations (their exact locations require to be identifiable, with hi-res grid refs;
2. sample locations - photos of the actual outcrops sampled;
3. geological correlation - the siltstone bands stated to match across headwall and RSF must be location-mapped, georeferenced, and presented in a standard geological section showing breadth and thickness of relevant exposures.

Photographic coverage - a Supplementary file of selected imagery would greatly help in comprehending this intriguing site and its setting (see eg. Wilson and Jarman 2022).  The writer will be providing an annotated Powerpoint slide set to the lead author with imagery from a site visit (28 May).

The writer's colleague in Lake District RSF studies, Peter Wilson, has now seen the paper, this review, and the Powerpoint slide set, and would encourage further development of this interesting paper along the lines suggested.

--- --- ---

**COMMENTARY 1  -  ASSUMPTIONS**

This paper boldly asserts (emphases by the writer) :
(647)
Thus, our hypothesis '*a steep, faulted, and unstable rock slope has experienced buttressing by glacial ice*' as proposed in the Introduction is corroborated here.

7.0 Conclusions

We have demonstrated that a RSF in the headwall of a cirque in the Lune gorge occurred as a slow downslope movement of an intact rock mass due to the presence of a supporting glacial ice mass buttressing the failed slope.

The Abstract is a little more circumspect :
The 18.0 ± 1.2 ka date is consistent with a small buttressing ice mass being present in the cirque at the time of regional deglaciation. The exposure age of 12.0 ± 0.8 ka represents a minimum age, as the highly-fractured surface of the failure plane has experienced post-failure mass-wasting. Considering the dates, it appears unlikely that the cirque was re-occupied by a substantial ice mass during the Younger Dryas Stadial.

The 'hypothesis' is a plausible, instructive, challenging, and valuable one.  However its 'proof' here rests on a number of important assumptions, which ought to be recognised.  Indeed, the entire paper could be recast as testing the converse 'null hypothesis' that

> 'if the feature in Great Coum is an arrested translational rockslide form of RSF, its emplacement could have occurred at or over any timescale in the later Quaternary and did not require the supportive presence of glacier ice'

(cf. the null hypothesis 'there are no rock glaciers in the British mountains, active or fossil' tested by Jarman et al 2013).

**THE ASSUMPTIONS**

**1 - the feature is an RSF.**

First, let us call this an Anomalous Terrain Feature (ATF), a term even less genetic than Discrete Debris Accumulation (DDA) coined by Brian Whalley.   This ATF is now included in the Wilson&Jarman Lake District Inventory as RSF #9.04, but without benefit of site visit, and with some reluctance.  It may look like a duck, but does it quack like a duck ?

Evidence for it being an RSF is almost entirely visual (circumstantial).  On walking over the site, the only clear positive indicator observed is an unusually fresh, jagged sub-vertical fracture-fissure (see photo).  Basal springs, a common indicator of RSF, were not evident at the immediate slope foot (even beneath the dry gully descended), but this was in a very dry spell;  OS 25k shows a group of smll watercourses emanating well below the talus zone inside the fenced enclosure, which may or may not be springs.

Against this being an RSF :

a.  the '**tread**' is a prominent bench or shelf, not unusually wide, such as commonly occurs on a cirque headwall, especially at mid-lower levels where curvature eases slope, if resistant bands create rocksteps.  Here, this may be a remnant of a wider rockstep.  In particular, the shelf fades laterally until buried in talus (especially northwards), whereas an RSF slipmass would have more 'emergence' from the slope.  It feels solid underfoot.
b.  the '**riser**' is actually stepped or tiered, in side profile very different from the rather sheer rim crags it is supposed to match;  it is unusual to be able to promenade rather

freely across an RSF slipmass on a steep slope, and to descend it easily top-to-foot;  it has an air of solidity saying 'in-situ bedrock' except for the unusual fracture.  In several places, large flakes have toppled from the mini-risers, revealing fresh faces with distinct in-situ feel, whereas in even a quasi-intact slipmass, deeper joint blocks would tend to come away because some internal rupturing and dilation must occur during translation (however firmly ice-buttressed).

c. there is no obvious **source configuration** for **slipmass restitution** to match (this is a common oversight with RSF misidentifications):  for the ATF to be an RSF requires either a planar source scar or an obtuse wedge-shaped cavity, neither of which can readily be demonstrated as restitutable (the writer here stops short of drawing diagrams showing fall-line trajectory in relation to ATF extent and rim topography, but is happy to consider proposals that would satisfy this key criterion).

d. there are almost no **dislocations** in the ATF, such as would commonly confirm a translated slipmass, especially of this travel distance, and even albeit this is a relatively small body.  Typically, the top surface might present a degree of back-tilting, with an upstand edge, whereas this one (even allowing for talus accumulation) rolls over weakly.  And the 'riser' might display antiscarp development, even on a sub-metric scale, as the emplaced mass dilates and disaggregates slightly (even if here, after 'debuttressing' rather than in transit).  The one fracture observed (and noted in the paper) is intriguing, but as the rockmass has not come apart materially, it betokens quasi-in situ rock slope deformation (RSD) - possibly a local rebound response - rather than a rockslide.

e. The ATF is split by a small central gully, rocky at the top becoming a grass chute (as descended), and by a much broader open swathe towards the north flank.  It is surprising that there is no obvious differential downslope movement facilitated by these lineaments, in such a long-travel mass, especially if it is held to have rotated laterally - where *en echelon* or mare's tail side-scarps have ben noted (eg. Sgurr na Iapaich Affric).

The writer generally rates RSFs for inventory purposes as definite-probable-possible, although Wilson and Jarman 2022 do not do this.  Here, Great Coum is not 'definite' on present evidence, and from field inspection remains nearer 'possible' than 'probable'.

**2 - geological parameters require the ATF to be an RSF**

Two features are invoked here:

a. lateral rotation, down-west - as revealed by the dip of the strata across the riser.  This does indeed seem convincing, both on imagery and viewed in the field.  However, there may a degree of optical illusion, compounding with the difficulty of reading 3-D structures presenting in 2-D.  The marked eastward 'dip' of the headwall to the east of the ATF is also reinforced visually by the elevational decline east along the cirque rim.  In fact, the ATF dip west looks broadly consistent with the headwall crag above, and with broken crags to the NW.  The paper refers to anticlinal structures, which may account for some of this effect.
   In any case, it is rather difficult to envisage how a slipmass of this bulk would rotate laterally while strongly ice-buttressed - its natural trajectory would be close to fall-line.  Examples of rockslide masses displaying such lateral rotation (as if a 'foundering ship') are hard to recall.

b. correlative strata - until the evidence for this is made available as requested above (which might clinch the RSF assumption) it can only be suggested that if these Silurian sand-silt deposits occur in rhythmic sequences, as quite often mentioned in geological literature, then it is possible that a visual impression of correlatable strata might be obtained from seeing such sequences within the headwall and riser that are actually 'pattern repeats'.

**3 - the RSF has a planar source configuration**

The paper assumes, for the purpose of Swedge modelling, a simple planar source (the 'basal failure plane' on Fig 4a, or 'head wall' as sample code HW implies). However, the full width of the tread as indicated in the paper extends well beyond the main north-facing crag, westwards below a broad grassy couloir (which the paper at one point hints to be the source), and indeed further west until below a degraded portion of the NE-facing cirque rim. Leaving aside the difficulty of fitting the ATF back to where it came from, inspection from vantage points along the rim down the fall-line suggests some form of obtuse wedge slide configuration. Clearly this has important consequences for modelling, as the slope angle of the wedge axis is less than that of the flanks, which as they converge must impede translation - one reason why so many RSFs are 'arrested'.

Here the writer has much pleasure in recommending the unpublished PhD of Graham Holmes (1984)
Holmes, G., 1984. Rock-slope Failure in Parts of the Scottish Highlands. Ph.D. thesis. University of Edinburgh (available online).
This was undertaken at the behest of Brian Sissons to demonstrate that RSFs associated strongly with his then- LLS limits (both of which proved to be wrong). 'Holmes' is now best known for its pioneering RSF inventory, including 'debris-free scarps' implying an earlier generation since glacially evacuated. His thorough exposition of the basic principles of slope stability in a British montane context is however crystal clear and superbly exemplified with field cases almost at lab-model scale. The effect of different source configurations is instructively set out (and complementary to the 'rock-toppling' PhD of Bob Watters 1972). His engineering geology methods would bear comparison with those adopted for this paper, including his measuring of a hundred joint aspects per site (which should be a thousand for serious projects !) to obtain spherical projections of joint sets available as slide surfaces.

Here at Great Coum, it can be observed that although perhaps not technically 'metamorphic' in the lack of crystallisation, the robust Silurian sediments have undergone sufficient modification to endow them with mutiple angular joint sets in addition to the bedding plane and any orthogonal jointing(see photos). This may also affect the modelling process.

**4 - the RSF has a planar sliding surface**

The paper assumes for Swedge modelling purposes a 'basal failure plane' which is essentially smooth and 2-D, with an allowance for a minor degree of 'waviness' suggested by slickensiding.

However, such ideal surfaces are increasingly becoming recognised as the exception, eg. in quartzite lithologies, or where through-going 'fault' discontinuities occur, perhaps lubricated with gouge. The paper alludes to thin partings of finer-grained sediment, which could assist in mobilisation, as do pelitic (micaceous) bands in coarse psammites in the Highlands. But the orientation of such partings would need to coincide closely with the inferred source configuration.

Generally though,
either a '**zone of crush**' is more likely to exist, as advocated for a major Lake District site
Jarman, D., Wilson, P., 2015a. Anomalous terrain at Dove Crags cirque–Gasgale Gill, English Lake District, interpreted as a large pre-LGM rock slope failure complex. Proc. Yorks. Geol. Soc. 60, 243–257.
Clear evidence for such zones of crush is now being found in borehole investigations of RSFs in Norway, including (spectacularly, rig helicoptered in) through the Mannen RSF, Romsda;

and/or a '**stepped basal configuration**' where a shearing of the rugosities is invoked to generate an effective sliding surface or zone.

Vick Bohme Rouyet Corner in Landslides 2020 depict a number of N Norway RSDs, with long-sections identifying both processes as proven by drilling.

Quite what impact these realities might have on the modelling outcome here is unclear, but it could potentially go either way. (The paper describes the slipmass as presenting 'friable bedrock' as if shaly, but evidence for this - extent, depth, mechanical strength, photos - is not provided. The bedrock exposures seen in touring the rim, the headwall with its coarse debris runs, and the ATF instead rather impressed with the general robustness and coherence of the bedrock.)

**4  -  the RSF is a wedge tapering to a pointed toe**

This assumption is central to the argument, implying that without restraining ice, such a wedge would coast downslope more freely than a blunter object - possibly even disintegrating into a rock avalanche.

The null hypothesis states that the putative RSF slipmass is a more conventional rectilinear (cuboidal) slab that has detached from the headwall on a weakness broadly parallel to its pre-failure surface as exposed in the rim. Such a slab would be impeded in translation by reducing headwall slope angle and by its blunt toe ploughing into the mid-lower slope deposits (till, talus, friable weathered bedrock).

To demonstrate that this is indeed a wedge toe would require geotechnical investigations, whether invasive or eg. GPR (see the work of Tim Davies at Clough Head, QRA Guide 2015). Viewed side-on though, the impression is of a solid rectilinear rockmass, which could either be a solid outcrop, or a rockslide mass ploughing into or progressively buried by talus etc.

**5  -  the RSF occurred entirely during the LGM and its deglaciation**

The null hypothesis states that the putative RSF could have initiated earlier in the Pleistocene, migrating incrementally to its present position, with or without ice support, as all other factors interacted.  For example, Great Coum might have been substantially excavated during the earlier cycles of cirque glaciation, and then tended to be dormant when suppressed beneath the great icesheet glaciations.  Given its low elevation and contraposed iceflow directions, continuing cirque glacier erosion of the headwall might then have been rather limited, allowing an RSF to evolve over multiple cycles.  In such scenarios, the whole issue of ice-supported translation becomes ever more complex and perhaps more of a secondary factor.

As an aside here, the writer once speculated that RSF could be initiated by the load of an icesheet several hundred metres thick (above summits) bearing on a fallible rim, thus a 'snap-off' like a boxcutter blade;  happily this never appeared in print.  Even so, it is an interesting thought experiment, when considering the rim of Great Coum and the ATF.  The writer subsequently doodled a cartoon of 'all the forces acting on a mountain slope, including during icesheet glaciation and after' and hawked it around a score of experts home and abroad, meeting with no dismissal, indeed positive interest, but with no-one volunteering to resource a simulation to put some numbers on the vectors.  This was in pursuit of the contention that RSF is primarily a rebound-driven response to Concentrated Erosion of Bedrock (see overview papers referenced).  Here, excavation of Great Coum would generate rebound stresses in the footslope developing into fracture systems migrating up to the rim.  But this usually envisages unusually rapid and recent cirque enlargement... unless the RSF is a slow-burner.

**COMMENTARY 2  -  GREAT COUM ANOMALIES**

The paper implicitly treats Great Coum as if it were a typical, representative Lake District RSF, from which conclusions of wider relevance for timing and failure mechanisms could be drawn.  This assumes that the putative RSF is not an anomalous 'outlier', to which we now turn.

**1 - anomalous cirque in location and elevation**

Great and Little Coum stand out on map and DEM as cirquefoms in a tract of intermediate hills where cirques are generally absent.  Ian Evans has two marginal cirques up Borrowdale nearby, as well as these.  That's all, westwards, until the great cirques of the High Street range.  Eastwards, the Howgill Fells - attaining a markedly higher prevailing elevation of 500-600m asl - entirely lack cirques except for the remarkably well developed Cautley locus.

The question must thus arise as to whether these are typical cirques, and why they might have originated at such a low elevation - even Cautley spans 650>200 m asl whereas Great Coum spans 470>150 m.  One possibility is that they have originated not as conventional cirques but from a trough-flank scallop as well seen nearby in Bannisdale and the upper Shap Fells Borrowdale, where bold arcuate escarpments on their mid-SW flanks ar conspicuous on imagery.  These little-studied 'nivation scars' (for want of knowing a more correct term - 'bananas' is suggested by a colleague) might suggest a transition from coldbased to warm-based ice coming off the low plateau and funnelling into the Kent-Lune discharge zone (compare Tweedsmuir and Dalveen, S Uplands).  The Coums together have a comparable NE aspect, albeit their setting is at the low end of the Lune gorge.  They could have initiated in this way, with segregation into clearer cirquefoms for local reasons.

**2 - anomalous RSF in zone of sparsity and in low elevation**

The Wilson&Jarman inventory depicts a broad lacuna in RSF incidence between the heads of Longsleddale/Haweswater and the eastern Howgill Fells, with Great Coum an isolated exception.  This is despite comparable 'available relief' in these valleys including the Shap Fells Borrowdale.

The source elevation of Great Coum RSF at 460 m asl is also relatively low. Although there are 13 cases of lesser elevation, three are compact rock avalanches from lower-level crags in Borrowdale near Derwent Water, three are on peripheral escarpments rather than in glaciated troughs and cirques, and five are in parafluvial contexts (see below).  This leaves for realistic comparison at such lower levels a small arrested rockslide on the Crummock rim of Mellbreak, and the remarkable Helm Crag RSF in the Grasmere trough, which is an RSD.

**3 - anomalous RSF-in-cirque**, in a Lake District context

There are 11 RSFs in cirque contexts in the Lake District inventory, of which interestingly only two are disintegrated rock avalanches - one within the oddly capacious and isolated Dead Crags cirque, Bakestall, north of Skiddaw, the other being the large and idiosyncratic Burtness Comb complex (see paper in submission by Wilson et al with cosmodates confirming the lower deposit as post-LGM and pre-LLS).  Of the three slope deformations (RSDs), two are trivial and one on the eastern spur of High Street (Caspel Gate) is a larger RSD of 0.09 km$^2$ on the Blea Water cirque flank as it extends into trough-head character.

Of the five arrested translational rockslides within cirques that might compare with the Great Coum case, four are very small (0.01-0.03 km$^2$) rimslips or 'rim nibbles' with high-level sources at 725-750 m asl, lowered by a few metres (Caudale Moor N, High Street NE, Black Sails, High Crag).  #8.01 Eller Peatpot on Black Combe is a slightly larger site of 0.05 km$^2$ which has a definite wedge lowered by ~10 m possibly nested within an earlier lowered berm and perhaps with a slide lobe into the cirque floor, subdued by cirque-glacial overriding.  It would repay investigation as falling between Burtness Comb and Great Coum in possible evolution.

In the Scottish Highlands, about 10-15% of RSFs are in cirque contexts, but a tabulation prepared in 2012 regrettably does not identify type or headscarp heights.  A brief search of possibly comparable areas such as Cowal does not yield comparable small long-travel intact rockslide slices from planar sources but doubtless they must exist.  Generally, headscarps in the 50-100 m+ height range are rare and associate with larger RSFs.

**4 - an anomalous context for an anomalous cirque and RSF**

Here, the coincidence between an anomalously isolated and low-level cirque, an anomalously isolated and atypical RSF, and the remarkable major landscape feature of the Lune gorge becomes compelling.

The writer regrets not being *au fait* with the longer-term evolution of the Lune gorge, but would guess that it is not a glacial breach, but perhaps an antecedent fluvial incision responding to uplift along the Lakes-Howgills axis[3]. If so, it will have undergone adaptation to accommodate ice discharge across this axis whenever the local ice divide was displaced northwards or southwards. As the paper states, there is now a U-profile with some shaving of the former interlocking spurs, although glacial trough development is still immature, hardly more so than its modest Borrowdale tributary. Indeed, at the south end it retains a more fluvial (V-form) character, perhaps due to diffluence through the Dillicar gap - see photos.

The unusual character of the Lune gorge has echoes along the Highland Boundary Fault, where a sequence of main valleys exiting the uplands across it are likewise not true breaches but have rather irregular, immaturely glaciated courses - and in several cases have clusters of RSFs upstream from them; also see the steep southern side of the S Uplands, where its small RSF clusters occur.

 It is thus possible that the continuing V>U conversion of the Lune gorge has been accompanied by RSF, within the classic glacial-paraglacial cycle - for which see
> Jarman D (2009): Paraglacial rock slope failure as an agent of glacial trough widening. In Knight, J. and Harrison, S. (eds). Periglacial and paraglacial processes and environments. Geological Society of London Special Publication 320, 103-131. doi:10.1144/SP320.8.

If additionally, the foot of the gorge has seen cyclical glacier advance, retreat and fluctuation, then repeated stressing and destressing of the valley sides will have occurred. RSF does seem to occur at such loci, as noted in the Eastern Pyrenees at trough-head transitions (TH-T), or here, at a transition from upland to lowland. Furthermore, the potential for RSF cavities to seed cirques has been noted since Clough (1896) and was recently explored by Ballantyne (2013).

Although the paper focusses on the ATF, dismissing other possible RSF indications around the GC-LC cirques as minor and not relevant, field inspection suggests that RSF may have been important in their evolution (see photos) :
> - the outer (eastern) corner of Great Coum has a distinct berm below the rim with a slightly protruding slope below, suggesting a short-travel rockslide, possibly ice-modified; it could have descended 30 m - assuming it is an RSF, of about 0.01 km$^2$;
> - along the rim above and west of this berm there are several minor steps and grooves suggestive of incipient rim failure (but not above the ATF - and note that the intricate dissection around Grayrigg Pike is not RSD but probably selective erosion by glacial meltwater outbreaks - see also the remarkable grooved terrain NE of the telecom mast);
* * *
[3] this detail is not mentioned by Rob Westaway (2009) in his advocacy of uplift of this axis since the mid-Pliocene

- ➢ the north-facing head of Little Coum is an anomalous planar slope with fatly swelling terrain below, hinting at a broad rockslide (slab-slide) with both source and debris ice-smoothed since; the rim has little wedge cavities in its angle;
- ➢ the short bold NE-facing crag in Little Coum looks like an RSF cavity of the debris-free type espoused by Ballantyne; there are very large angular blocks possibly of this origin on the outer apron;
- ➢ the outer rim of Little Coum has recesses and lineaments suggestive of sub-RSF dislocation and slippage, perhaps ice-smoothed
- ➢ the apron is elevated above that of Great Coum, and the pronounced step-down between them is marked by a curious linear scarp and berm, seemingly in bedrock; it could result from selective stripping by ice along conducive joints, or just possibly could be failed, thus a headscarp and berm to a lowered slice.
- ➢ around the corner into Borrowdale there is a distinct cavity and slipmass, if sub-RSF in scale.

All this might suggest that the GC-LC compound cirque could have originated not in the conventional way (whatever that is, a matter perhaps rather glossed over, but presumably exploiting fluvial valley heads with conducive aspect and concavity), but from one or two significant cavities created by proto-RSFs at this focus of CEB and slope stressing.

**COMMENTARY 3  -  FURTHER ISSUES**

If this - already admirable and thought-provoking - paper is to have wider value and applicability to sites beyond that studied, some further issues arise.

**1  -  RSF modes and contexts**

Firstly, it is stated that  "the slope would have failed catastrophically, if not supported by glacial ice" (30-31).  A corollary might be that all the cataclasmic RSFs in the Lake District were unsupported by ice, and thus postdate final local deglaciation, which is not unreasonable.  This only deals with the minority that are cataclasmic - 15 of 84 :  conversely it could be taken to imply that the 27 translational rockslides would have collapsed without ice support.  Of course their cavities, failure surfaces, and geology would all differ, but it would be useful to know just how sensitive Swedge - or other analytical techniques - might be to such parameters.

Then there is the neglected fact that a significant minority of RSFs occur in non-glaciated valleys - almost 10% in the Lake District, and a majority in mid-Wales and most of the Southern Uplands (and of course the totality in non-glaciated ranges abroad).  Here, they are typically in fluvial (V-form) side-valleys envisaged to have undergone rapid incision or deepening by peak meltwater discharges during deglaciation, with consequent slope destabilisation.  They are thus termed fluvial RSFs, or 'parafluvial' if they are not responding directly to ordinary fluvial erosion at the slope foot, but are on fluvially steepened slopes.  Of the nine Lake District cases, most are translational rockslides, little different in size range and form from their paraglacial counterparts.  As they cannot have been supported by

glacier ice, and despite some being lowered well down the slope, some other process for initiating and then arresting them must be found (one practical civil engineer simply responded 'they dried up' ) - and if this applies to all parafluvial RSFs, it will doubtless apply to some paraglacial ones.

**2 - scalability**

The 'Holmes model' of a quartzite block on a smooth tilted joint plane can readily be envisaged, both at its Peak Friction Angle while joint-cemented and at its Residual Friction Angle while unrestrained - and it can be seen at a scale of say 100x100x10 m (thus just qualifying as an RSF) to have moved quasi-intact in places as diverse as Glen Dessary, Glen Quoich, and Jura.[4]

However, if the Holmes block is scaled up progressively, towards RSFs of average size (~0.20 km$^2$) and beyond, it can be imagined as becoming both ever stickier, as the ideal sliding surface becomes 'noisier', and ever less able to remain quasi-intact, as its internal inhomogeneities proliferate and respond to the stresses of gravity and translation and so forth.  This is why, we envisage, long-travelled quasi-intact RSFs are rare.

And that is without considering glacier ice support.  Here, the downwasting of the ice cannot be expected to be smooth and uniform, nor can the slipmass be assumed to remain so closely in contact as to glide down the failure plane.  They must surely play catch-up, with phases of ice wasting and stillstand, and of slipmass pausing and remobilising.  Each cycle must expose the slipmass to internal and external stresses rendering it more liable to sticking fast - or vulnerable to progressive or calamitous disintegration.  And the larger the slipmass, the more prone.

Thus while this paper offers a fascinating analysis under near-ideal conditions at field-experiment scale, it must be wondered if the glacier-ice support scenario can usefully be scaled up to the generality of RSF.

Here the debuttressing argument comes to the fore.  From conversations with both Tim Davies and Sam McColl, the writer sees the logic of their reasoning - developed of course in respect of very large RSDs in alpine New Zealand troughs - that glaciers are plastic and thus also deformable, and that a failed slope can sag extensively even while in contact with the glacier, and remain in that metastable position after ice withdrawal.  The writer has not refreshed and updated the course of this debate, but recalls that this 'debunking of debuttressing' may have been rowed back from to some degree.

Its relevance here, in considering scalability, is that it seems intuitively more reasonable for a large glacier to hold a small RSF in place, controlling its translation, than for a waning glacier to restrain and control a large RSF.  If so, the less scalable are the findings of this
* * *
[4] Indeed a scale model of that has been played with, on slabs at differerent inclinations, to the writer's great satisfaction if sore arms.  And demonstrated to a student on the slopes of Ben Vorlich, with a convenient slab at the 'tipping point' where an applied fingertip obtained translationshe was mapping the RSFs in her study area with no idea that they could have moved on any combination of joint sets, not just the foliation surface  - and went on to join a firm of engineering consultants.

paper, and the more likely it is that medium and larger RSFs have not been ice supported, and could thus be synchronous with or later than final deglaciation.

**3 - the single date issue**

Recourse to the journal website and its record of 'submission-in-progress' (a novelty for this writer) indicates that the single-date issue has led to a previous rejection and has already been taken up by the Associate Editor as requiring greater circumspection. The Supplementary files address this issue, but add little to the simple fact that 'dates cost money' especially when much bulkier samples are required to obtain datable material than with the usual quartz-knob method (the writer has assisted CK Ballantyne in homing in on such quartz-knobs, and has also sampled Dartmoor tors for conveniently back-packable flake sizes - but RSFs are rarely in quartz-rich granite alas).

The writer has discussed with Derek Fabel the vexed question of number of samples required for reliable dating. He suggested, ideally, collecting ten, processing the first five, and if statistically consistent, calling a halt (if not, carry on). Given the cost, he accepted that four or even three might give a close-enough approximation to the actual age. Ballantyne has published a set of rock avalanche ages for the Highlands and Ireland based on three dates per site, with the protocol that if two are close and one an outlier, that can be rejected (IS Evans confirms that this is statistically invalid, as the single date could be 'more right' than the pair).

Here, the author admits that a single date is not really adequate to give a reliable age, and that the headscarp date may be little more than a limiting age. However the writer has further concerns, which would persist even with (say) three dates (and even once the precise locations of the two single samples were provided and inspected) :

 ➢ the source area is not a simple, visible rock scar of homogenous form or character. As discussed above, it is unclear whence the putative slipmass has come, but several sampling points across the 100-200 m width of bold crag - grassy bay (does it possess outcrops ?) - weaker crag would be desirable;
 ➢ the 'riser' is likewise a staircase of treads and risers, of varying boldness laterally and vertically, which should be sampled to reflect average conditions;
 ➢ there has been considerable wastage along these mini-risers, as mentioned, and it may be difficult to judge what facet(s) truly represent the unmodified post-ice surface, thus several sampling points are needed.

Ideally, it would be intriguing to sample vertically down both source scar and riser, to see if they 'young' downwards with progressive slipmass displacement and ice surface lowering[5]; the thick scatter of coarse angular blocks on the riser could be dated, to give an earliest age for its separation from the rim to reveal the extant crags.
* * *
[5] this was proposed by the writer for the Dartmoor tor dating project, where a tor had been quarried into, thus revealing a profile through the former ground surface and on to a depth below cosmo-penetration;  the principle was accepted...

Possibly Schmidt hammer dating would give affordable insights into some of these concerns, as conducted for the Burtness Comb UD-LD by Peter Wilson with some success.

In this light, the two dates currently available deserve mention as broadly supportive, with all due caveats, and as advancing the case for more systematic dating of this site and others to compare or contrast.

---

## Referee Comment (RC2)

- 1 Ice buttressing-controlled rock slope failure on a cirque headwall,
- 2 English Lake District
- 3
- 4 Paul A. Carlinga, b, John D. Jansenc, Teng Sud, e, Jane Lund Andersenf, Mads Faurschou
- 5 Knudsenf
- 6 a Geography and Environmental Science, University of Southampton, Southampton, SO17
- 7 1BJ, UK.
- 8 b Lancaster Environment Centre, Lancaster University, Bailrigg, Lancaster, LA1 4YW, UK.
- 9 c GFÚ Institute of Geophysics, Czech Academy of Sciences, Prague, Czechia.
- d University of Chinese Academy of Sciences, Beijing 100049, China.
- 11 e Laboratory of Water Cycle and Related Land Surface Processes, Institute of Geographical
- 12 Sciences and Natural Resources Research, Chinese Academy of Sciences, Beijing, 100101,
- 13 China.
- 14 f Department of Geoscience, Aarhus University, Aarhus, Denmark.
- 15
- 16 Corresponding author: Paul A. Carling (p.a.carling@soton.ac.uk)
- 17
- 18 Key Points
- 19 Geometry and mechanics of cirque rock slope failure defined from the local geology
- 20 Rock slope failed slowly due to ice buttressing the cirque headwall
- 21 Rock slope failure occurred during deglaciation
- 22

**23 Abstract**

| 25after the Last Glacial Maximum, but controls and timing of failures remain poorly known. A26cirque headwall failure was investigated to determine failure mechanisms and timing. The27translated wedge of rock is thin and lies on a steep failure plane, yet the friable strata were28not disrupted by downslope movement. Fault lines and a failure surface, defining the29wedge, were used as input to a numerical model of rock wedge stability. Various failure30scenarios indicated that the slope would have failed catastrophically, if not supported by31glacial ice in the base of the cirque. The amount of ice required to buttress the slope is32insubstantial, indicating likely failure during thinning of the cirque glacier. We propose that,33as the ice thinned, the wedge was lowered slowly down the cirque headwall gradually34exposing the failure plane. A cosmogenic 10 Be surface exposure age of 18.0 ± 1.2 ka from35the outer surface of the wedge indicates Late Devensian de-icing of the back wall of the36cirque, with a second exposure age from the upper portion of the failure plane yielding 12.037± 0.8 ka. The 18.0 ± 1.2 ka date is consistent with a small buttressing ice mass being present38in the cirque at the time of regional deglaciation. The exposure age of 12.0 ± 0.8 ka39represents a minimum age, as the highly-fractured surface of the failure plane has40experienced post-failure mass-wasting. Considering the dates, it appears unlikely that the41cirque was re-occupied by a substantial ice mass during the Younger Dryas Stadial.                                              | 24 | Rock slope failures in the English Lake District have been associated with deglacial processes         |
|-------------------------------------------------------------------------------------------------------------------------------------------------------------------------------------------------------------------------------------------------------------------------------------------------------------------------------------------------------------------------------------------------------------------------------------------------------------------------------------------------------------------------------------------------------------------------------------------------------------------------------------------------------------------------------------------------------------------------------------------------------------------------------------------------------------------------------------------------------------------------------------------------------------------------------------------------------------------------------------------------------------------------------------------------------------------------------------------------------------------------------------------------------------------------------------------------------------------------------------------------------------------------------------------------------------------------------------------------------------------------------------------------------------------------------------------------------------------------------------------------------------------------------------------------------------------------------------------------------------------------------------------------------------|----|--------------------------------------------------------------------------------------------------------|
|  <li>cirque headwall failure was investigated to determine failure mechanisms and timing. The</li> <li>translated wedge of rock is thin and lies on a steep failure plane, yet the friable strata were</li> <li>not disrupted by downslope movement. Fault lines and a failure surface, defining the</li> <li>wedge, were used as input to a numerical model of rock wedge stability. Various failure</li> <li>scenarios indicated that the slope would have failed catastrophically, if not supported by</li> <li>glacial ice in the base of the cirque. The amount of ice required to buttress the slope is</li> <li>insubstantial, indicating likely failure during thinning of the cirque glacier. We propose that,</li> <li>as the ice thinned, the wedge was lowered slowly down the cirque headwall gradually</li> <li>exposing the failure plane. A cosmogenic 10Be surface exposure age of 18.0 ± 1.2 ka from</li> <li>the outer surface of the wedge indicates Late Devensian de-icing of the back wall of the</li> <li>cirque, with a second exposure age from the upper portion of the failure plane yielding 12.0</li> <li>± 0.8 ka. The 18.0 ± 1.2 ka date is consistent with a small buttressing ice mass being present</li> <li>in the cirque at the time of regional deglaciation. The exposure age of 12.0 ± 0.8 ka</li> <li>represents a minimum age, as the highly-fractured surface of the failure plane has</li> <li>experienced post-failure mass-wasting. Considering the dates, it appears unlikely that the</li> <li>cirque was re-occupied by a substantial ice mass during the Younger Dryas Stadial.</li>  | 25 | after the Last Glacial Maximum, but controls and timing of failures remain poorly known. A             |
|  <li>translated wedge of rock is thin and lies on a steep failure plane, yet the friable strata were</li> <li>not disrupted by downslope movement. Fault lines and a failure surface, defining the</li> <li>wedge, were used as input to a numerical model of rock wedge stability. Various failure</li> <li>scenarios indicated that the slope would have failed catastrophically, if not supported by</li> <li>glacial ice in the base of the cirque. The amount of ice required to buttress the slope is</li> <li>insubstantial, indicating likely failure during thinning of the cirque glacier. We propose that,</li> <li>as the ice thinned, the wedge was lowered slowly down the cirque headwall gradually</li> <li>exposing the failure plane. A cosmogenic 10Be surface exposure age of 18.0 ± 1.2 ka from</li> <li>the outer surface of the wedge indicates Late Devensian de-icing of the back wall of the</li> <li>cirque, with a second exposure age from the upper portion of the failure plane yielding 12.0</li> <li>± 0.8 ka. The 18.0 ± 1.2 ka date is consistent with a small buttressing ice mass being present</li> <li>in the cirque at the time of regional deglaciation. The exposure age of 12.0 ± 0.8 ka</li> <li>represents a minimum age, as the highly-fractured surface of the failure plane has</li> <li>experienced post-failure mass-wasting. Considering the dates, it appears unlikely that the</li> <li>cirque was re-occupied by a substantial ice mass during the Younger Dryas Stadial.</li>                                                                                                    | 26 | cirque headwall failure was investigated to determine failure mechanisms and timing. The               |
|  <li>not disrupted by downslope movement. Fault lines and a failure surface, defining the</li> <li>wedge, were used as input to a numerical model of rock wedge stability. Various failure</li> <li>scenarios indicated that the slope would have failed catastrophically, if not supported by</li> <li>glacial ice in the base of the cirque. The amount of ice required to buttress the slope is</li> <li>insubstantial, indicating likely failure during thinning of the cirque glacier. We propose that,</li> <li>as the ice thinned, the wedge was lowered slowly down the cirque headwall gradually</li> <li>exposing the failure plane. A cosmogenic 10Be surface exposure age of 18.0 ± 1.2 ka from</li> <li>the outer surface of the wedge indicates Late Devensian de-icing of the back wall of the</li> <li>cirque, with a second exposure age from the upper portion of the failure plane yielding 12.0</li> <li>± 0.8 ka. The 18.0 ± 1.2 ka date is consistent with a small buttressing ice mass being present</li> <li>in the cirque at the time of regional deglaciation. The exposure age of 12.0 ± 0.8 ka</li> <li>represents a minimum age, as the highly-fractured surface of the failure plane has</li> <li>experienced post-failure mass-wasting. Considering the dates, it appears unlikely that the</li> <li>cirque was re-occupied by a substantial ice mass during the Younger Dryas Stadial.</li>                                                                                                                                                                                                             | 27 | translated wedge of rock is thin and lies on a steep failure plane, yet the friable strata were        |
|  <li>wedge, were used as input to a numerical model of rock wedge stability. Various failure</li> <li>scenarios indicated that the slope would have failed catastrophically, if not supported by</li> <li>glacial ice in the base of the cirque. The amount of ice required to buttress the slope is</li> <li>insubstantial, indicating likely failure during thinning of the cirque glacier. We propose that,</li> <li>as the ice thinned, the wedge was lowered slowly down the cirque headwall gradually</li> <li>exposing the failure plane. A cosmogenic 10Be surface exposure age of 18.0 ± 1.2 ka from</li> <li>the outer surface of the wedge indicates Late Devensian de-icing of the back wall of the</li> <li>cirque, with a second exposure age from the upper portion of the failure plane yielding 12.0</li> <li>± 0.8 ka. The 18.0 ± 1.2 ka date is consistent with a small buttressing ice mass being present</li> <li>in the cirque at the time of regional deglaciation. The exposure age of 12.0 ± 0.8 ka</li> <li>represents a minimum age, as the highly-fractured surface of the failure plane has</li> <li>experienced post-failure mass-wasting. Considering the dates, it appears unlikely that the</li> <li>cirque was re-occupied by a substantial ice mass during the Younger Dryas Stadial.</li>                                                                                                                                                                                                                                                                                                           | 28 | not disrupted by downslope movement. Fault lines and a failure surface, defining the                   |
|  <li>scenarios indicated that the slope would have failed catastrophically, if not supported by</li> <li>glacial ice in the base of the cirque. The amount of ice required to buttress the slope is</li> <li>insubstantial, indicating likely failure during thinning of the cirque glacier. We propose that,</li> <li>as the ice thinned, the wedge was lowered slowly down the cirque headwall gradually</li> <li>exposing the failure plane. A cosmogenic 10Be surface exposure age of 18.0 ± 1.2 ka from</li> <li>the outer surface of the wedge indicates Late Devensian de-icing of the back wall of the</li> <li>cirque, with a second exposure age from the upper portion of the failure plane yielding 12.0</li> <li>± 0.8 ka. The 18.0 ± 1.2 ka date is consistent with a small buttressing ice mass being present</li> <li>in the cirque at the time of regional deglaciation. The exposure age of 12.0 ± 0.8 ka</li> <li>represents a minimum age, as the highly-fractured surface of the failure plane has</li> <li>experienced post-failure mass-wasting. Considering the dates, it appears unlikely that the</li> <li>cirque was re-occupied by a substantial ice mass during the Younger Dryas Stadial.</li>                                                                                                                                                                                                                                                                                                                                                                                                            | 29 | wedge, were used as input to a numerical model of rock wedge stability. Various failure                |
|  <li>glacial ice in the base of the cirque. The amount of ice required to buttress the slope is</li> <li>insubstantial, indicating likely failure during thinning of the cirque glacier. We propose that,</li> <li>as the ice thinned, the wedge was lowered slowly down the cirque headwall gradually</li> <li>exposing the failure plane. A cosmogenic 10Be surface exposure age of 18.0 ± 1.2 ka from</li> <li>the outer surface of the wedge indicates Late Devensian de-icing of the back wall of the</li> <li>cirque, with a second exposure age from the upper portion of the failure plane yielding 12.0</li> <li>± 0.8 ka. The 18.0 ± 1.2 ka date is consistent with a small buttressing ice mass being present</li> <li>in the cirque at the time of regional deglaciation. The exposure age of 12.0 ± 0.8 ka</li> <li>represents a minimum age, as the highly-fractured surface of the failure plane has</li> <li>experienced post-failure mass-wasting. Considering the dates, it appears unlikely that the</li> <li>cirque was re-occupied by a substantial ice mass during the Younger Dryas Stadial.</li>                                                                                                                                                                                                                                                                                                                                                                                                                                                                                                                | 30 | scenarios indicated that the slope would have failed catastrophically, if not supported by             |
|  <li>insubstantial, indicating likely failure during thinning of the cirque glacier. We propose that,</li> <li>as the ice thinned, the wedge was lowered slowly down the cirque headwall gradually</li> <li>exposing the failure plane. A cosmogenic 10Be surface exposure age of 18.0 ± 1.2 ka from</li> <li>the outer surface of the wedge indicates Late Devensian de-icing of the back wall of the</li> <li>cirque, with a second exposure age from the upper portion of the failure plane yielding 12.0</li> <li>± 0.8 ka. The 18.0 ± 1.2 ka date is consistent with a small buttressing ice mass being present</li> <li>in the cirque at the time of regional deglaciation. The exposure age of 12.0 ± 0.8 ka</li> <li>represents a minimum age, as the highly-fractured surface of the failure plane has</li> <li>experienced post-failure mass-wasting. Considering the dates, it appears unlikely that the</li> <li>cirque was re-occupied by a substantial ice mass during the Younger Dryas Stadial.</li>                                                                                                                                                                                                                                                                                                                                                                                                                                                                                                                                                                                                                    | 31 | glacial ice in the base of the cirque. The amount of ice required to buttress the slope is             |
|  <li>as the ice thinned, the wedge was lowered slowly down the cirque headwall gradually</li> <li>exposing the failure plane. A cosmogenic 10Be surface exposure age of 18.0 ± 1.2 ka from</li> <li>the outer surface of the wedge indicates Late Devensian de-icing of the back wall of the</li> <li>cirque, with a second exposure age from the upper portion of the failure plane yielding 12.0</li> <li>± 0.8 ka. The 18.0 ± 1.2 ka date is consistent with a small buttressing ice mass being present</li> <li>in the cirque at the time of regional deglaciation. The exposure age of 12.0 ± 0.8 ka</li> <li>represents a minimum age, as the highly-fractured surface of the failure plane has</li> <li>experienced post-failure mass-wasting. Considering the dates, it appears unlikely that the</li> <li>cirque was re-occupied by a substantial ice mass during the Younger Dryas Stadial.</li>                                                                                                                                                                                                                                                                                                                                                                                                                                                                                                                                                                                                                                                                                                                              | 32 | insubstantial, indicating likely failure during thinning of the cirque glacier. We propose that,       |
|  <li>exposing the failure plane. A cosmogenic 10Be surface exposure age of 18.0 ± 1.2 ka from</li> <li>
[revised manuscript text omitted]

87

---

## Referee Comment (RC3)

[referee-annotated manuscript omitted]

---

## Author Comment (AC1)

**Carling et al**
**Ice buttressing-controlled rock slope failure on a cirque headwall, English Lake District**

**Earth Surface Dynamics 2023**

**This document is the authors response to comments RC1 and RC2.**
**The authors' replies to the commentary provided by the reviewer are in light blue**

Reviewer comments

**David Jarman**
mountain landform research
Scotland

SUMMARY
**Recommendation** - accept with minor changes, subject to attention to the figures
(with scope for more substantial revision/improvement at editorial/authorial discretion)
This paper addresses an apparently trivial landform feature, a very small and unimpressive RSF, which (most unusually, in the British mountains and generally) happens to be discernible from a major road (M6). It analyses the slope stresses acting on such a slipped mass by an engineering geology technique Swedge, with and without support from glacier ice. It dates the outer face and the scar above, and from their significantly older and younger ages, it states that this corroborates a contention that the slipped mass could only have been emplaced thus with the support of a waning valley glacier.
The paper is in fact of great interest, and a novel contribution to geomorphology, for four main reasons: 1. case studies of individual RSFs in the British mountains are uncommon, and those analysing their engineering geology are vanishingly rare (and essentially confined to a couple of PhD theses on Scottish Highlands sites, now decades old). 2. few RSFs in the Lake District (two) and Highlands (a score or so) have been dated - all of them fully disintegrated rock avalanches, whereas this is an arrested translational slide. Even globally, quasi-intact rockslides have seldom been dated due to obvious sampling difficulties, with the emphasis on 'antiscarp' trench faces rather than outward-facing slopes as here. 3. glacier support for RSF emplacement is not an entirely original notion, but to the writer's incomplete knowledge this may be a pioneering study of a possible demonstration example, in Britain at least. 4. morphologically, RSFs in cirques form a small minority, with some of those on outer flanks rather than headwalls; and the distance travelled by the putative slipmass while remaining quasi-intact - 110 m - is remarkable, especially for such a small feature. (the authors are perhaps too modest here, and could increase the impact of the paper by placing its originality more firmly within the British and global montane RSF literature).

We thank Dr Jarman for his detailed and enthusiastic review of our submission. In particular, he indicates that the manuscript is of great interest and might be accepted subject to minor changes, involving attention to the figures. He provides four main reasons why the submission is a novel contribution to geomorphology. In brief, 1) the approach

using 'engineering geology' is to be applauded; 2) we have dated our RSF which is unusual; 3) the suggestion of glacial support for a RSF, although not original, is here for the first time demonstrated as a possibility; 4) the distance travelled by the RSF is remarkable given the bedrock remained intact. It is for these reasons that we prepared the manuscript bringing a numerical stability approach to the subject of a RSF that seems to have descended remarkably intact from a steep failure plane and date the event.

The reviewer also concludes that 'the two dates currently available deserve mention as broadly supportive, with all due caveats, and as advancing the case for more systematic dating of this site and others to compare or contrast'.

The reviewer provides 15 pages of commentary, some of which is focussed on positioning this study into the broader context of RSFs and other slope failures within the Lake District and potentially further afield. Below we provide comment and have revised the manuscript where appropriate, but we have not replied in detail to those points that lie outside of the scope of the current submission. It is evident that the reviewer, in his enthusiasm, would like us to extend our study to make generalised statements that apply to other RSFs. We purposefully have not done so, as we believe that a larger corpus of case studies on the controls on RSFs is required before such generalizations can be made. Moreover, to extend the manuscript content and figure would lead to a submission in excess of the guidance on manuscript lengths.

Although the site is trivial in size - at 0.03 km2 in extent by a standard method it is only just above the threshold of 0.01 km2 for designation as RSF (see 'References') - this proves not to be a demerit, for it is in effect a scale model or field-laboratory experiment. Although this site is unusual in some ways, it is not *sui generis*, and the findings are relevant to the wider study of RSF behaviour.  Again, these virtues could be made clearer.

Although the RSF is small, it is not trivial in-as-much as it offered the opportunity to explore the failure mechanisms within a numerical framework, which is novel, as indicated by the reviewer.  We purposely did not over-emphasize the possibility of extending our conclusions more widely, as we acknowledge that such a model may not apply to many other RSFs.

While the paper is eminently publishable essentially as it stands, it relies on a number of assumptions, stated and unstated, which bear closer scrutiny. The paper could be greatly improved by adumbrating these assumptions; alternatively, the editors and authors may be well content with 'putting it out there' as an aunt sally to provoke and stimulate further debate and work. If the latter course is followed, it is recommended that at least the 'corroborated hypothesis' be expressed more circumspectly[1], with due acknowledgement that it does rest on a set of convenient assumptions, as befits a controlled experiment.

We address the issue of assumptions below at the point where the reviewer details his comments fully.

1 here it is noted that the paper has previously ben rejected because of its reliance on two single dates, and that the Associate Editor has already advised circumspection in this regard. It might help further if the emphasis is taken away from reliance on these dates, and the case study presented as a multi-pronged assessment, with the dates providing a measure of comfort, but far from essential to the conclusions reached.

Before exploring these assumptions in the COMMENTARY that follows, some practicalities can be dealt with.

REFERENCES

The lead author is now aware of a new overview paper on Lake District RSF, published as the present paper was being submitted :

Peter Wilson & David Jarman (2022) Rock slope failure in the Lake District,
NW England: an overview, Geografiska Annaler: Series A, Physical Geography, 104:3, 201-225,
DOI: 10.1080/04353676.2022.2120261

In addition, the reference to Jarman (2005) should be 2006; this paper is largely superseded by an overview paper:

Jarman D, Harrison S (2019): Rock slope failure in the British mountains. Geomorphology 340, 202-233.

These two papers do not require any major alterations in the present paper, beyond updating the regional data, and perhaps noting that the RSF conceptual typology has been generalised since 2005/6 for much wider montane applicability. They do contain discussions of RSF timing and causation which may be of interest.

TEXT

The submitted paper is returned with 'sticky notes' for possible clarification, and marking a few trivial edits. They also identify passages discussed in the Commentary below.

We have addressed all the minor comments and 'trivial edits' shown on the marked manuscript. We have added the additional number of RSF in the Lake District supported by the references provided by the referee. We thank the referee for these references.

FIGURES

It is recommended that the visual presentation of the site be substantially upgraded:

Fig 1 the arrow is too crude and impinges on the feature - it would be more instructive to delineate the identified RSF site boundary, including 'cavity' or source area at rim, and basal extent, with a fine dotted line, and a fine arrow suggesting slipmass trajectory.

This fig. presents an overview of the Great Coum cirque shown without any labels to obscure the morphology (labels and lines are added in a slightly more close-up version presented in Fig. 4B). The N-arrow is clear and does not impinge upon any landforms discussed in this MS. The RSF is delineated as described by the reviewer in Fig. 4B.

Fig 2 this regional map is neither sufficiently regional nor adequately detailed at Lune gorge context scale. Two maps are needed:
◻ regional, including
◻ full named extents of Shap Fells and Howgill Fells, with Lake District as far west as say High Street;

It is not clear what would be gained by showing all of the Shap Fells and Howgill Fells in this figure. Those sites are not directly relevant to the focus of our MS, which is the RSF at Great Coum.

-- full extent of upper Lune basin, with general ice movements as long curvilinears, not crude arrows, and famous drumlin field just to SW identified and placed in ice outflow context;

The linear arrows clearly indicate ice flow direction. We intentionally generalised the ice trajectories in linear segments because adding curvilinear shapes would be unsupported speculation. The drumlin field is not relevant to our study.

-- ice movements not just coded but numbered 1-2-3 to clarify sequence (nb. present caption omits ST2; it could helpfully specify what all three codes represent)

Good suggestion. We have modified the arrow labels to indicate the temporal sequence and edited the caption accordingly.

-- extent of Silurian outcrop, with adjacent lithologies;

It is not clear what a lithological map would add. This MS does not aim to analyse the regional geology. A very large map would be required to include other lithologies as the region of Great Coum is all Silurian rocks.

-- RSFs, with small-large sizes, thus emphasising the High Street cluster and sparsity here (easily obtained from Wilson&Jarman 20222);

This MS sets out to examine the Great Coum cirque including a detailed engineering-based failure analysis. Other RSFs in the region are beyond the scope of this MS.

-- glacial cirques (from Ian Evans inventory), perhaps with grades/elevations, again emphasising sparsity around here and unusually low elevation of GC/LC;

Beyond the scope of this MS. We report the elevation of the ELA in the GC/LC area.

-- the M6 even !!

The meaning of this comment is difficult to decipher. The position of the M6 highway is not relevant to the study.

2 four additional small sites have since been identified from imagery in this area - see appendix
-- local, perhaps as broad as present but less N-S to zoom in closer in landscape format, including
-- named highest summits and elevations in each hill mass (Grayrigg Pike, not Forest, being the peak closest to the site)
-- elevations along Lune gorge floor and rims

None of this information is directly relevant to our study.

-- RSFs, distinguishing parafluvial cases in Howgill V-shape valleys (#9.03 A/B) from paraglacial locations as here and Cautley #9.01/02
-- cirques

This information is not related to the present study. We cannot agree to most of these suggestions. Figure 2 is a location map which is kept clear of unnecessary information, as stated above. It serves adequately to locate the study site. The reviewers' suggestions relate to placing the RSF into a much wider context which is not the focus of the submission.

Fig 3 these long profiles really require a proper map of the two cirques to locate them, which could very usefully also depict the glacial moraine features etc described - this might be includable on the 'local context map' requested above.

It is not necessary. These long profiles are short and self-explanatory; they run along the centrelines of the cirques as described in the manuscript. The caption to Fig. 3 includes the GPS end points of the long profiles such that the reader can easily find these on platforms such as Google Earth.  The till and moraines mentioned in the main text provide context but are not required to develop the model, so we purposefully did not add this unnecessary detail to any figure.

Fig 4A the schematic wedge already gives one actual dimension, the 44º slope - why not add the other dimensions ?
(nb. the word 'debris' in nigh invisible)

It is important that definition diagrams allow the reader to see the essential detail alone. This schematic does this without the clutter of additional dimensions which are detailed in the main text and within Table 2. We have redrawn the figure to ensure the word 'debris' is clear.

Fig 4B we are brought very little closer to visualising the RSF than in Fig 1, and with the same GEarth image offering no change in perspective - a different angle / historical image, or a photo, would add considerably to our grasp.

We chose to use exactly the same image within Fig. 4B as in Fig. 1, so the reader can see the RSF without clutter in Fig. 1 and then see it again with the minimum additional necessary notation in Fig. 4B that allows the model to be established. Fig. 4B contains the essential information the reviewer noted was missing in Fig. 1.

(nb 1. it is not clear what the three components marked "c" intend - is a mappable fault trace actually observed to be displaced down the headwall, or exposed in it at a lower level by removal of the slipmass, or is this merely suggestive ?)

The text explains that the fault line defined by 'c' can be seen in the field and was mapped. It appears the reviewer has reviewed the figures without referring to the main text.

(nb 2. sample codes HW and OSF are stated in the caption and should be added to the figure) - ideally, with the actual dates (as concise ~ages) to aid our grasping that they are older above younger - the writer has to recheck this every time....

Good point. The sample locations for cosmogenic dating are shown in Fig. 4B. The notation HW and OSF are now added. The caption and the main text allow each one to be identified. We do not add the actual dates to this figure because these are not reported until a later section.

Fig 7 this useful diagram is very hard to decipher from the caption. Do we understand there to be two pentahedral volumes:
- potential, delineated by A-D (can it not be extended E-F for greater clarity ?)
- needed, toned purple
The term 'pentahedral volume' should be stated as 'potential pentahedral volume' at its first appearance in the caption, and might be toned lightly.

The caption has been revised as suggested. The potential pentahedral volume has not been lightly toned as the geometry is already clear due to the annotation A, B, C and D. Further tones would obscure the figure. For clarity, we only labelled those points required to define a pentahedral, so no further labels have been added.

The 'needed pentahedral volume' should be stated to be purple (why does its contact with the riser need to be curvy, in a schematic ?)

The volume has now been labelled as 'purple' in the caption. The curvy line is to indicate that the ice was unlikely to lie against the riser is a uniform manner. We have added a sentence to the caption to make this clear.

RECOMMENDED ADDITIONAL FIGURES
1. detailed site map - this map of Great Coum would define the RSF extent and components properly, in a way which the thumbnail sketch in Fig 4B cannot. It would mark and codename the two sample locations (their exact locations require to be identifiable, with hi-res grid refs;

We have explained above why we have not adopted this suggestion.

2. sample locations - photos of the actual outcrops sampled;

A good suggestion. We have added photographs to the Supplement.

3. geological correlation - the siltstone bands stated to match across headwall and RSF must be location-mapped, georeferenced, and presented in a standard geological section showing breadth and thickness of relevant exposures.

In our opinion a detailed site map is not required given the focus of the submission. The dimensions of the components of RSF are given in Table 2. The locations of the samples are already provided in Fig. 4B and the GPS locations are given in Table 2. A detailed stratigraphical log is not required to establish the model. Such an exercise would constitute a new study, leading to substantial stratigraphical information beyond the requirements of this submission that would warrant an additional publication. The site has never been stratigraphically mapped by any party (including BGS), as key exposures are not accessible except by abseil on unsafe rock faces. Correlation with outcrops at distance is not possible.

Photographic coverage - a Supplementary file of selected imagery would greatly help in comprehending this intriguing site and its setting (see eg. Wilson and Jarman 2022). The writer will be providing an annotated Powerpoint slide set to the lead author with imagery from a site visit (28 May).

Good suggestion, we have added a selection of photos to the Supp.

The writer's colleague in Lake District RSF studies, Peter Wilson, has now seen the paper, this review, and the Powerpoint slide set, and would encourage further development of this interesting paper along the lines suggested.

Reviewer's commentary is supposedly confidential until the Editor agrees to post it online.

In his enthusiasm the reviewer requests so much detail that the manuscript would far exceed the required submission length.

--- --- ---
**COMMENTARY 1 - ASSUMPTIONS**
This paper boldly asserts (emphases by the writer) :
(647)
Thus, our hypothesis '*a steep, faulted, and unstable rock slope has experienced buttressing by glacial ice*' as proposed in the Introduction is corroborated here.
We have demonstrated that a RSF in the headwall of a cirque in the Lune gorge occurred as a slow downslope movement of an intact rock mass due to the presence of a supporting glacial ice mass buttressing the failed slope.
The Abstract is a little more circumspect :
The 18.0 ± 1.2 ka date is consistent with a small buttressing ice mass being present in the cirque at the time of regional deglaciation. The exposure age of 12.0 ± 0.8 ka represents a minimum age, as the highly-fractured surface of the failure plane has experienced post-failure mass-wasting. Considering the dates, it appears unlikely that the cirque was re-occupied by a substantial ice mass during the Younger Dryas Stadial.
The 'hypothesis' is a plausible, instructive, challenging, and valuable one. However its 'proof' here rests on a number of important assumptions, which ought to be recognised. Indeed, the entire paper could be recast as testing the converse 'null hypothesis' that
'if the feature in Great Coum is an arrested translational rockslide form of RSF, its emplacement could have occurred at or over any timescale in the later Quaternary and did not require the supportive presence of glacier ice'

(cf. the null hypothesis 'there are no rock glaciers in the British mountains, active or fossil' tested by Jarman et al 2013).

We are aware that a null hypothesis could have been applied. The suggestion offered by the reviewer is a compound hypothesis largely focussing on the timing of the failure. Our study is not primarily focussed on the timing although timing is an important consideration. The issue of 'assumptions' is addressed below.

**THE ASSUMPTIONS**
**1 - the feature is an RSF.**
First, let us call this an Anomalous Terrain Feature (ATF), a term even less genetic than Discrete Debris Accumulation (DDA) coined by Brian Whalley. This ATF is now included in the Wilson&Jarman Lake District Inventory as RSF #9.04, but without benefit of site visit, and with some reluctance. It may look like a duck, but does it quack like a duck ?
Evidence for it being an RSF is almost entirely visual (circumstantial). On walking over the site, the only clear positive indicator observed is an unusually fresh, jagged sub-vertical fracture-fissure (see photo). Basal springs, a common indicator of RSF, were not evident at the immediate slope foot (even beneath the dry gully descended), but this was in a very dry spell; OS 25k shows a group of smll watercourses emanating well below the talus zone inside the fenced enclosure, which may or may not be springs.

It is unclear just how much of the site was visited by the reviewer and how much time was spent on site. Evidence for the RSF is not 'almost entirely visual' as suggested by the reviewer. As was included in the original submission, as well as the vertical fissure (noted by the reviewer) there is also the evidence of three fault planes providing lateral constraint to the rock slope wedge as well as a bounding basal fault plane down which the wedge descended. The occurrence of pale silt bands in the RSF was carefully matched to the occurrence of the same silts bands in the headwall which is how the distance travelled downslope was derived. The dip of the strata within the RSF as well as within the headwall was measured to derive the small plunge. All this detail was included in our initial submission. Basal springs are present at the location but given the unusual drought we have witnessed this year no water has been issuing from these risings when the reviewer visited the site.

Against this being an RSF :
  a.  the '**tread**' is a prominent bench or shelf, not unusually wide, such as commonly occurs on a cirque headwall, especially at mid-lower levels where curvature eases slope, if resistant bands create rocksteps. Here, this may be a remnant of a wider rockstep. In particular, the shelf fades laterally until buried in talus (especially northwards), whereas an RSF slipmass would have more 'emergence' from the slope. It feels solid underfoot.

      As noted in the original submission the silt bands within the RSF match the silt bands within the headwall, so there is no possibility that the rock mass is a shelf. The lateral extent of the tread is partly buried by talus which has accumulated from small rock falls from the headwall. Any large rock mass will feel solid underfoot, so we are unsure what the comment about solid footfall is meant to convey.

b. the '**riser**' is actually stepped or tiered, in side profile very different from the rather sheer rim crags it is supposed to match; it is unusual to be able to promenade rather freely across an RSF slipmass on a steep slope, and to descend it easily top-to-foot; it has an air of solidity saying 'in-situ bedrock' except for the unusual fracture. In several places, large flakes have toppled from the mini-risers, revealing fresh faces with distinct in-situ feel, whereas in even a quasi-intact slipmass, deeper joint blocks would tend to come away because some internal rupturing and dilation must occur during translation (however firmly ice-buttressed).

The riser unusually is **not stepped** and this is clearly seen from the southerly aspect. A modest degree of stepping occurs close by the fissure that the reviewer states that he descended. The strata are disrupted to a very minor degree and plunge 2$^{\circ}$ steeper than the headwall strata. The joints may have been introduced by minor dilation but equally these can be primary as was noted in the original manuscript. The limited degree of break-up of the slipped wedge despite the thinness of the beds and the steep failure plane is the first thing we noted as unusual about this RSF.

c. there is no obvious **source configuration** for **slipmass restitution** to match (this is a common oversight with RSF misidentifications): for the ATF to be an RSF requires either a planar source scar or an obtuse wedge-shaped cavity, neither of which can readily be demonstrated as restitutable (the writer here stops short of drawing diagrams showing fall-line trajectory in relation to ATF extent and rim topography, but is happy to consider proposals that would satisfy this key criterion).

We are surprised the reviewer makes this comment seeing he has visited the site. The original manuscript mentions the source area at lines 234-237 but the source area is clearly seen, especially when viewed from across the valley. We state in the original submission that the 'The RSF caused headwall retreat in the vicinity of the present grassy slope, leaving the intact steep rocky sections of the backwall to either side,'.

d. there are almost no **dislocations** in the ATF, such as would commonly confirm a translated slipmass, especially of this travel distance, and even albeit this is a relatively small body. Typically, the top surface might present a degree of back-tilting, with an upstand edge, whereas this one (even allowing for talus accumulation) rolls over weakly. And the 'riser' might display antiscarp development, even on a sub-metric scale, as the emplaced mass dilates and disaggregates slightly (even if here, after 'debuttressing' rather than in transit). The one fracture observed (and noted in the paper) is intriguing, but as the rockmass has not come apart materially, it betokens quasi-in situ rock slope deformation (RSD) - possibly a local rebound response - rather than a rockslide.

The lack of major dislocation, as noted above, is the prime reason we suspected a slow descent for the RSF. The top surface (the tread) does locally show back tilting *i.e.*, antiscarp behaviour but some of the tread is covered by scree. The large vertical fissure is aligned with a BGS-mapped fault line that extends back into the headwall

and beyond on the plateau above. The faulting in the parent geology might include post-glacial rebound but otherwise predates the Quaternary, as is evident from the references cited in the original submission – "Moseley (1968; 1972) considered the considerable complexity of the regional structure and noted folding, steep discontinuous local faulting, joint patterns". Without information on local rebound we do not wish to enter into speculation as to the role of rebound other than to note the effect of isostatic controls which we had done in the submission at line 729.

e. The ATF is split by a small central gully, rocky at the top becoming a grass chute (as descended), and by a much broader open swathe towards the north flank. It is surprising that there is no obvious differential downslope movement facilitated by these lineaments, in such a long-travel mass, especially if it is held to have rotated laterally - where *en echelon* or mare's tail side-scarps have been noted (eg. Sgurr na lapaich Affric).

In our original submission, we report differential downslope movement of 2° from field measurement of plunge and this difference was an emergent property of one of our simulations wherein the NW side of the RSF descends further downslope that the SW side of the failure.

The writer generally rates RSFs for inventory purposes as definite-probable-possible, although Wilson and Jarman 2022 do not do this. Here, Great Coum is not 'definite' on present evidence, and from field inspection remains nearer 'possible' than 'probable'.

This comment does not relate to the content of the manuscript.

**2 - geological parameters require the ATF to be an RSF**
Two features are invoked here:
a. lateral rotation, down-west - as revealed by the dip of the strata across the riser. This does indeed seem convincing, both on imagery and viewed in the field. However, there may a degree of optical illusion, compounding with the difficulty of reading 3-D structures presenting in 2-D. The marked eastward 'dip' of the headwall to the east of the ATF is also reinforced visually by the elevational decline east along the cirque rim. In fact, the ATF dip west looks broadly consistent with the headwall crag above, and with broken crags to the NW. The paper refers to anticlinal structures, which may account for some of this effect.

We note that the reviewer finds that the plunge of the strata across the RSF is 'convincing'. 'Eyeballing' the plunge of the RSF and the plunge within the headwall and is clearly not adequate to reach a conclusion related to a small angle of plunge. Rather, the reported plunge is not due to inspection of images but due to measurements of plunge in the field. The elevation changes along the top of the cirque are due to regional structural constraints which are not relevant to the issue of local plunge within the RSF.

In any case, it is rather difficult to envisage how a slipmass of this bulk would rotate laterally while strongly ice-buttressed - its natural trajectory would be close to fall-line. Examples of rockslide masses displaying such lateral rotation (as if a 'foundering ship') are hard to recall.

It is not clear why the reviewer comes to this conclusion. The modelling we present indicates that the loading of the ice mass across the riser of the RSF can be variable so there is no issue with the RSF descending further down to the NW than the SE. We did not pursue this issue due to the 'infinite' ice loading possibilities, rather we decided to present a simple case as an example that is convincing. Adding complexity, such as varied ice loadings, goes beyond the capacity of the modelling and more especially goes beyond the constraints of the data that can be derived from field study. Such detail would be seen as 'going too far' in the view of many reviewers and is a step too far in the opinion of the current authors.

   b. correlative strata - until the evidence for this is made available as requested above (which might clinch the RSF assumption) it can only be suggested that if these Silurian sand-silt deposits occur in rhythmic sequences, as quite often mentioned in geological literature, then it is possible that a visual impression of correlatable strata might be obtained from seeing such sequences within the headwall and riser that are actually 'pattern repeats'.

We kept the manuscript succinct. As evidence, we reported that pale silt bands found within the headwall are also found in similar position within the RSF and that bed thicknesses within the RSF do not correlate with bed thicknesses in the undisturbed outcrops to either side of the RSF. The level of detail requested by the reviewer goes beyond the focus of the manuscript, which is a slope stability exercise. The complexity of the Silurian succession would be the province of another publication altogether and difficult to complete due to the danger in accessing all the cliff face exposures.

**3 - the RSF has a planar source configuration**
The paper assumes, for the purpose of Swedge modelling, a simple planar source (the 'basal failure plane' on Fig 4a, or 'head wall' as sample code HW implies). However, the full width of the tread as indicated in the paper extends well beyond the main north-facing crag, westwards below a broad grassy couloir (which the paper at one point hints to be the source), and indeed further west until below a degraded portion of the NE-facing cirque rim. Leaving aside the difficulty of fitting the ATF back to where it came from, inspection from vantage points along the rim down the fall-line suggests some form of obtuse wedge slide configuration. Clearly this has important consequences for modelling, as the slope angle of the wedge axis is less than that of the flanks, which as they converge must impede translation - one reason why so many RSFs are 'arrested'.

The paper makes no assumption about a planar failure plane. As explain in the original main text, the failure plane is exposed at several locations and the plane extends both to the north (as noted by the reviewer) but also the south and the angle of this plane was measured. This information was included in the original manuscript. There is no difficulty in fitting the RSF wedge back into its source location as the pale silt bands allow for this to be done in terms of altitudinal adjustment. The headwall is now much degraded by post-glacial weathering – hence the scree – so a 'perfect' fit into a source alcove is not possible,

*although such an alcove exists to the NW of the RSF as was stated in the original submission at lines 235- 237.*

Here the writer has much pleasure in recommending the unpublished PhD of Graham Holmes (1984)
Holmes, G., 1984. Rock-slope Failure in Parts of the Scottish Highlands. Ph.D. thesis. University
of Edinburgh (available online).
This was undertaken at the behest of Brian Sissons to demonstrate that RSFs associated strongly with his then- LLS limits (both of which proved to be wrong). 'Holmes' is now best known for its pioneering RSF inventory, including 'debris-free scarps' implying an earlier generation since glacially evacuated. His thorough exposition of the basic principles of slope stability in a British montane context is however crystal clear and superbly exemplified with field cases almost at lab-model scale. The effect of different source configurations is instructively set out (and complementary to the 'rock-toppling' PhD of Bob Watters 1972). His engineering geology methods would bear comparison with those adopted for this paper, including his measuring of a hundred joint aspects per site (which should be a thousand for serious projects !) to obtain spherical projections of joint sets available as slide surfaces.

*Thank you for these references but to keep our submission focussed we have not extended the text in such a way as to require their inclusion. As detailed in the original submission, our RSF occurred on a fault plane and not along a joint set.*

Here at Great Coum, it can be observed that although perhaps not technically 'metamorphic' in the lack of crystallisation, the robust Silurian sediments have undergone sufficient modification to endow them with mutiple angular joint sets in addition to the bedding plane and any orthogonal jointing (see photos). This may also affect the modelling process.

*In the original submission, we reported the presence of joints in the RSF and within the source bedrock. However, it is not clear how such jointing might be included in a failure model when the slipped mass appears to be coherent down a failure plane defined by a fault line. In our opinion it is unlikely that a more sophisticated model than the one we present would add deeper insight to the failure mechanisms, indeed such a model would venture into speculation unsupported by data.*

**4 - the RSF has a planar sliding surface**

The paper assumes for Swedge modelling purposes a 'basal failure plane' which is essentially smooth and 2-D, with an allowance for a minor degree of 'waviness' suggested by slickensiding. However, such ideal surfaces are increasingly becoming recognised as the exception, eg. in quartzite lithologies, or where through-going 'fault' discontinuities occur, perhaps lubricated with gouge. The paper alludes to thin partings of finer-grained sediment, which could assist in mobilisation, as do pelitic (micaceous) bands in coarse psammites in the Highlands. But the orientation of such partings would need to coincide closely with the inferred source configuration.

We do not assume a basal planar surface, rather one is exposed locally on which we measured the angle. The limited exposures show this to be smooth. The site may be an exception but that does not mean it lacks interest. Quite the converse, the simplicity of the geometry makes it ideal for investigation and modelling. The pale beds are not thin partings as the reviewer would 'wish', and we make no suggestion that they could assist in mobilization as the observed orientations are not concordant with the failure plane.

Generally though,
either a '**zone of crush**' is more likely to exist, as advocated for a major Lake District site Jarman, D., Wilson, P., 2015a. Anomalous terrain at Dove Crags cirque–Gasgale Gill, English Lake District, interpreted as a large pre-LGM rock slope failure complex. Proc. Yorks. Geol. Soc. 60, 243–257.
Clear evidence for such zones of crush is now being found in borehole investigations of RSFs in Norway, including (spectacularly, rig helicoptered in) through the Mannen RSF, Romsda; and/or a '**stepped basal configuration**' where a shearing of the rugosities is invoked to generate an effective sliding surface or zone.
Vick Bohme Rouyet Corner in Landslides 2020 depict a number of N Norway RSDs, with long-sections identifying both processes as proven by drilling.
Quite what impact these realities might have on the modelling outcome here is unclear, but it could potentially go either way. (The paper describes the slipmass as presenting 'friable bedrock' as if shaly, but evidence for this - extent, depth, mechanical strength, photos - is not provided. The bedrock exposures seen in touring the rim, the headwall with its coarse debris runs, and the ATF instead rather impressed with the general robustness and coherence of the bedrock.)

The information presented here by the reviewer is very interesting but not relevant to the focus of the manuscript. As noted above, the exposed portions of the basal failure plane indicate failure along a pre-existing fault line. Other than a small amount of post-glacial spalling on the failure plane surface, there is no field evidence of a major crush zone and one would not be expected given the thinness of the failed rock wedge. We do not report the details of the mechanical strength of the Silurian beds, as this information is not required for our modelling framework. Rather we report that the frequency of joints and the presence of cleavage as this would lead one to expect an unsupported RSF to break up during descent. The level of information already supplied is sufficient for the purposes of the project. The headwall is heavily fractured and incompetent, as is witnessed from viewing the headwalls from below and as is evident in extensive scree development.

**4 - the RSF is a wedge tapering to a pointed toe**
This assumption is central to the argument, implying that without restraining ice, such a wedge would coast downslope more freely than a blunter object - possibly even disintegrating into a rock avalanche.
The null hypothesis states that the putative RSF slipmass is a more conventional rectilinear (cuboidal) slab that has detached from the headwall on a weakness broadly parallel to its pre-failure surface as exposed in the rim. Such a slab would be impeded in translation by reducing headwall slope angle and by its blunt toe ploughing into the mid-lower slope deposits (till, talus, friable weathered bedrock).

To demonstrate that this is indeed a wedge toe would require geotechnical investigations, whether invasive or eg. GPR (see the work of Tim Davies at Clough Head, QRA Guide 2015). Viewed side-on though, the impression is of a solid rectilinear rockmass, which could either be a solid outcrop, or a rockslide mass ploughing into or progressively buried by talus etc.

The wedge-shape is not an assumption. It is evident that the reviewer did not visit the SE side of the RSF where the wedge shape is clearly evident. There is no reason for the reviewer to state that a conventional failure is cuboidal, rather the literature shows all kinds of shapes for RSFs. There is no talus at the foot of the wedge which could have buried the toe.

**5 - the RSF occurred entirely during the LGM and its deglaciation**

The null hypothesis states that the putative RSF could have initiated earlier in the Pleistocene, migrating incrementally to its present position, with or without ice support, as all other factors interacted. For example, Great Coum might have been substantially excavated during the earlier cycles of cirque glaciation, and then tended to be dormant when suppressed beneath the great icesheet glaciations. Given its low elevation and contraposed iceflow directions, continuing cirque glacier erosion of the headwall might then have been rather limited, allowing an RSF to evolve over multiple cycles. In such scenarios, the whole issue of ice-supported translation becomes ever more complex and perhaps more of a secondary factor.

As an aside here, the writer once speculated that RSF could be initiated by the load of an icesheet several hundred metres thick (above summits) bearing on a fallible rim, thus a 'snap-off' like a boxcutter blade; happily this never appeared in print. Even so, it is an interesting thought experiment, when considering the rim of Great Coum and the ATF. The writer subsequently doodled a cartoon of 'all the forces acting on a mountain slope, including during icesheet glaciation and after' and hawked it around a score of experts home and abroad, meeting with no dismissal, indeed positive interest, but with no-one volunteering to resource a simulation to put some numbers on the vectors. This was in pursuit of the contention that RSF is primarily a rebound-driven response to Concentrated Erosion of Bedrock (see overview papers referenced). Here, excavation of Great Coum would generate rebound stresses in the footslope developing into fracture systems migrating up to the rim. But this usually envisages unusually rapid and recent cirque enlargement... unless the RSF is a slow-burner.

We do not suggest that the RSF could not have been initiated earlier within the Pleistocene. Such scenarios are legion, rather the thinness of the intact strata within the thin wedge leads the authors to conclude the RSF would not survive repeated cycles of glaciation and down slope motion. Rather we adopt Occam's Razor and conclude that the most likely scenario is a late glacial slow descent into the base of the cirque. Rebound stresses may well have led to the release of the wedge of rock, but it is bounded on all sides by faults which are pre-Quaternary (see references cited in the original submission).

**COMMENTARY 2 - GREAT COUM ANOMALIES**

The paper implicitly treats Great Coum as if it were a typical, representative Lake District RSF, from which conclusions of wider relevance for timing and failure mechanisms could be drawn.

We must take exception to this remark.  Nowhere within the manuscript is it made explicit or implicit that the site is typical and representative of Lake District RSFs. We would not wish to make such an argument.  This erroneous assumption colours the comments of the reviewer below.

This assumes that the putative RSF is not an anomalous 'outlier', to which we now turn.
**1 - anomalous cirque in location and elevation**
Great and Little Coum stand out on map and DEM as cirquefoms in a tract of intermediate hills where cirques are generally absent. Ian Evans has two marginal cirques up Borrowdale nearby, as well as these. That's all, westwards, until the great cirques of the High Street range. Eastwards, the Howgill Fells - attaining a markedly higher prevailing elevation of 500-600m asl - entirely lack cirques except for the remarkably well developed Cautley locus. The question must thus arise as to whether these are typical cirques, and why they might have originated at such a low elevation - even Cautley spans 650>200 m asl whereas Great Coum spans 470>150 m. One possibility is that they have originated not as conventional cirques but from a trough-flank scallop as well seen nearby in Bannisdale and the upper Shap Fells Borrowdale, where bold arcuate escarpments on their mid-SW flanks ar conspicuous on imagery. These little-studied 'nivation scars' (for want of knowing a more correct term - 'bananas' is suggested by a colleague) might suggest a transition from cold-based to warm-based ice coming off the low plateau and funnelling into the Kent-Lune discharge zone (compare Tweedsmuir and Dalveen, S Uplands). The Coums together have a comparable NE aspect, albeit their setting is at the low end of the Lune gorge. They could have initiated in this way, with segregation into clearer cirque forms for local reasons.

The origin of the cirque forms is not germane to the focus of the manuscript.  The two hillside hollows have been recognized as cirques by many authors (Marr and Fearnsides, 1909; Moulson, 1966; King, 1976; Barr et al., 2017; Clark et al., 2018) and included in the BRITICE compendium. Harvey (2017) speculated that Great and Little Coums were only nivation hollows developed late in the Devensian, but these features are likely too large to have been produced by snowpatch processes and qualify as Type 3 or 4 cirques in the Evans and Cox (1995) classification.

**2 - anomalous RSF in zone of sparsity and in low elevation**
The Wilson & Jarman inventory depicts a broad lacuna in RSF incidence between the heads of Longsleddale/Haweswater and the eastern Howgill Fells, with Great Coum an isolated exception. This is despite comparable 'available relief' in these valleys including the Shap Fells Borrowdale.
The source elevation of Great Coum RSF at 460 m asl is also relatively low. Although there are 13 cases of lesser elevation, three are compact rock avalanches from lower-level crags in Borrowdale near Derwent Water, three are on peripheral escarpments rather than in glaciated troughs and cirques, and five are in parafluvial contexts (see below). This leaves for realistic comparison at such lower levels a small arrested rockslide on the Crummock rim of Mellbreak, and the remarkable Helm Crag RSF in the Grasmere trough, which is an RSD.

This statement supports the supposition that there well might well be a RSF in Great Coum. There is no issue to be addressed within the manuscript with reference to the referee's statement here.

**3 - anomalous RSF-in-cirque**, in a Lake District context
There are 11 RSFs in cirque contexts in the Lake District inventory, of which interestingly only two are disintegrated rock avalanches - one within the oddly capacious and isolated Dead Crags cirque, Bakestall, north of Skiddaw, the other being the large and idiosyncratic Burtness Comb complex (see paper in submission by Wilson et al with cos modates confirming the lower deposit as post-LGM and pre-LLS). Of the three slope deformations (RSDs), two are trivial and one on the eastern spur of High Street (Caspel Gate) is a larger RSD of 0.09 km2 on the Blea Water cirque flank as it extends into trough-head character. Of the five arrested translational rockslides within cirques that might compare with the Great Coum case, four are very small (0.01-0.03 km2) rimslips or 'rim nibbles' with high-level sources at 725-750 m asl, lowered by a few metres (Caudale Moor N, High Street NE, Black Sails, High Crag). #8.01 Eller Peatpot on Black Combe is a slightly larger site of 0.05 km2 which has a definite wedge lowered by ~10 m possibly nested within an earlier lowered berm and perhaps with a slide lobe into the cirque floor, subdued by cirque-glacial overriding. It would repay investigation as falling between Burtness Comb and Great Coum in possible evolution.
In the Scottish Highlands, about 10-15% of RSFs are in cirque contexts, but a tabulation prepared in 2012 regrettably does not identify type or headscarp heights. A brief search of possibly comparable areas such as Cowal does not yield comparable small long-travel intact rockslide slices from planar sources but doubtless they must exist. Generally, headscarps in the 50-100 m+ height range are rare and associate with larger RSFs.

This information is interesting but does not reflect on any statements we have made in the manuscript.

**4 - an anomalous context for an anomalous cirque and RSF**

Here, the coincidence between an anomalously isolated and low-level cirque, an anomalously isolated and atypical RSF, and the remarkable major landscape feature of the Lune gorge becomes compelling.

It is not clear from the reviewer's text developed above why the reviewer should consider the RSF in Great Coum to be anomalous. From the text above supplied by the reviewer it seems there are a variety of rock slope failure types across a spectrum of altitudes and environmental conditions.

The writer regrets not being *au fait* with the longer-term evolution of the Lune gorge, but would guess that it is not a glacial breach, but perhaps an antecedent fluvial incision responding to uplift along the Lakes-Howgills axis3. If so, it will have undergone adaptation to accommodate ice discharge across this axis whenever the local ice divide was displaced northwards or southwards. As the paper states, there is now a U-profile with some shaving

of the former interlocking spurs, although glacial trough development is still immature, hardly more so than its modest Borrowdale tributary. Indeed, at the south end it retains a more fluvial (V-form) character, perhaps due to diffluence through the Dillicar gap - see photos.

The issues raised here are not directly related to the content of the manuscript. For the reviewer's information all these points have been addressed in Carling et al., 2023, *Proceedings of the Geologists' Association*, 134, 139-165.

3 this detail is not mentioned by Rob Westaway (2009) in his advocacy of uplift of this axis since the mid-Pliocene
The unusual character of the Lune gorge has echoes along the Highland Boundary Fault, where a sequence of main valleys exiting the uplands across it are likewise not true breaches but have rather irregular, immaturely glaciated courses - and in several cases have clusters of RSFs upstream from them; also see the steep southern side of the S Uplands, where its small RSF clusters occur.
It is thus possible that the continuing V>U conversion of the Lune gorge has been accompanied by RSF, within the classic glacial-paraglacial cycle - for which see
Jarman D (2009): Paraglacial rock slope failure as an agent of glacial trough widening. In Knight, J. and Harrison, S. (eds). Periglacial and paraglacial processes and environments. Geological Society of London Special Publication 320, 103-131. doi:10.1144/SP320.8.

Once again, this comment does not relate to the content of the manuscript.  However, we agree that it is possible that the RSFs may play a role in widening of glacial troughs.  However, the Great Coum RSF is the only recognized RSF in the Lune Gorge, such that such failures do not seem to be important in the local context.  We do not wish to extend the manuscript into realms of speculation for which we do not have data in support.

If additionally, the foot of the gorge has seen cyclical glacier advance, retreat and fluctuation, then repeated stressing and destressing of the valley sides will have occurred. RSF does seem to occur at such loci, as noted in the Eastern Pyrenees at trough-head transitions (TH-T), or here, at a transition from upland to lowland. Furthermore, the potential for RSF cavities to seed cirques has been noted since Clough (1896) and was recently explored by Ballantyne (2013).

It should be evident from the existing manuscript that the 'particular' geometry of faults within the vicinity of the Great Coum headwall have conditioned the actual geometry of the RSF in Great Coum.  We do not wish to draw any inference that the RSF in Great Coum is a major influence, or otherwise, on the development of that cirque.  Nor, having only one example, would we wish to suggest that RSF are important in the development of cirques more widely.

Although the paper focusses on the ATF, dismissing other possible RSF indications around the GC-LC cirques as minor and not relevant, field inspection suggests that RSF may have been important in their evolution (see photos) :

-- the outer (eastern) corner of Great Coum has a distinct berm below the rim with a slightly protruding slope below, suggesting a short-travel rockslide, possibly ice-modified; it could have descended 30 m - assuming it is an RSF, of about 0.01 km2;
-- along the rim above and west of this berm there are several minor steps and grooves suggestive of incipient rim failure (but not above the ATF - and note that the intricate dissection around Grayrigg Pike is not RSD but probably selective erosion by glacial meltwater outbreaks - see also the remarkable grooved terrain NE of the telecom mast);

We are aware of these additional features which is why we make passing reference to these very minor mass failures in the manuscript. The grooving is structurally controlled, and we do not speculate on the origin as it has nothing to do with the focus of the manuscript. Other than indicating that other very minor rock falls and RSFs have occurred in Great Coum to either side of the RSF we have modelled, the information regarding these features is not relevant to the modelling and interpretation within our manuscript.

-- the north-facing head of Little Coum is an anomalous planar slope with fatly swelling terrain below, hinting at a broad rockslide (slab-slide) with both source and debris ice-smoothed since; the rim has little wedge cavities in its angle;
-- the short bold NE-facing crag in Little Coum looks like an RSF cavity of the debris-free type espoused by Ballantyne; there are very large angular blocks possibly of this origin on the outer apron;
-- the outer rim of Little Coum has recesses and lineaments suggestive of sub-RSF dislocation and slippage, perhaps ice-smoothed
-- the apron is elevated above that of Great Coum, and the pronounced step-down between them is marked by a curious linear scarp and berm, seemingly in bedrock; it could result from selective stripping by ice along conducive joints, or just possibly could be failed, thus a headscarp and berm to a lowered slice.
-- around the corner into Borrowdale there is a distinct cavity and slipmass, if sub-RSF in scale.

We are aware already of these details which are not relevant to the RSF in Great Coum.

All this might suggest that the GC-LC compound cirque could have originated not in the conventional way (whatever that is, a matter perhaps rather glossed over, but presumably exploiting fluvial valley heads with conducive aspect and concavity), but from one or two significant cavities created by proto-RSFs at this focus of CEB and slope stressing.

The origin of the two cirques is not the focus of the manuscript. Little Coum is certainly structurally controlled.

**COMMENTARY 3 - FURTHER ISSUES**
If this - already admirable and thought-provoking - paper is to have wider value and applicability to sites beyond that studied, some further issues arise.

Thank you for noting that the manuscript is admirable and thought provoking. We only wish to promote further research and constructive argument as to the mechanisms and the timing of RSF.

**1 - RSF modes and contexts**

Firstly, it is stated that "the slope would have failed catastrophically, if not supported by glacial ice" (30-31). A corollary might be that all the cataclasmic RSFs in the Lake District were unsupported by ice, and thus postdate final local deglaciation, which is not unreasonable. This only deals with the minority that are cataclasmic - 15 of 84 : conversely it could be taken to imply that the 27 translational rockslides would have collapsed without ice support. Of course their cavities, failure surfaces, and geology would all differ, but it would be useful to know just how sensitive Swedge - or other analytical techniques - might be to such parameters.

The reviewer is attempting to extend our findings more widely to apply to many RSFs for which we do not have data to consider their failure modes. The purpose of the manuscript is to promote such considerations, but we purposefully make no claims that our model can be applied elsewhere.  However, we do encourage others to apply numerical stability models where this is possible.

Then there is the neglected fact that a significant minority of RSFs occur in non-glaciated valleys - almost 10% in the Lake District, and a majority in mid-Wales and most of the Southern Uplands (and of course the totality in non-glaciated ranges abroad). Here, they are typically in fluvial (V-form) side-valleys envisaged to have undergone rapid incision or deepening by peak meltwater discharges during deglaciation, with consequent slope destabilisation. They are thus termed fluvial RSFs, or 'parafluvial' if they are not responding directly to ordinary fluvial erosion at the slope foot, but are on fluvially steepened slopes. Of the nine Lake District cases, most are translational rockslides, little different in size range and form from their paraglacial counterparts. As they cannot have been supported by glacier ice, and despite some being lowered well down the slope, some other process for initiating and then arresting them must be found (one practical civil engineer simply responded 'they dried up' ) - and if this applies to all parafluvial RSFs, it will doubtless apply to some paraglacial ones.

We acknowledge that there are RSFs which are 'non-glacial'. Our manuscript does not address these.

**2 - scalability**

The 'Holmes model' of a quartzite block on a smooth tilted joint plane can readily be envisaged, both at its Peak Friction Angle while joint-cemented and at its Residual Friction Angle while unrestrained - and it can be seen at a scale of say 100x100x10 m (thus just qualifying as an RSF) to have moved quasi-intact in places as diverse as Glen Dessary, Glen Quoich, and Jura.4

4 Indeed a scale model of that has been played with, on slabs at differerent inclinations, to the writer's great satisfaction if sore arms. And demonstrated to a student on the slopes of Ben Vorlich, with a convenient slab at the 'tipping point' where an applied fingertip obtained translationshe was mapping the RSFs in her study area with no idea that they could have moved on any combination of joint sets, not just the foliation surface - and went on to join a firm of engineering consultants.

However, if the Holmes block is scaled up progressively, towards RSFs of average size (~0.20 km2) and beyond, it can be imagined as becoming both ever stickier, as the ideal sliding surface becomes 'noisier', and ever less able to remain quasi-intact, as its internal inhomogeneities proliferate and respond to the stresses of gravity and translation and so forth. This is why, we envisage, long-travelled quasi-intact RSFs are rare.

We appreciate that there have been other attempts to model RSFs making various generalized assumptions. Our model applies specifically to a pentahedral wedge of rock. The key point is that despite examining thousands of potential RSF geometries that are plausible within Great Coum, the result remains the same: the wedge cannot have been stable and would descend rapidly if not supported.

And that is without considering glacier ice support. Here, the downwasting of the ice cannot be expected to be smooth and uniform, nor can the slipmass be assumed to remain so closely in contact as to glide down the failure plane. They must surely play catch-up, with phases of ice wasting and stillstand, and of slipmass pausing and remobilising. Each cycle must expose the slipmass to internal and external stresses rendering it more liable to sticking fast - or vulnerable to progressive or calamitous disintegration. And the larger the slipmass, the more prone.

We acknowledge the lengthy time period associated with deglaciation and that the RSF may have descended in stop:start motion within ice over an unknown time period. We address the controls on such motion in the existing manuscript from lines 544 onwards whereby we consider in principle changing the ice load distribution, the presence of a bergschrund and lubrication; all of which are related to deglaciation. We also note the effects of weathering and brittle fracture that can lead to the RSF being stable or liable to downward motion. An extended period for lowering the rock wedge within an ice mass would likely lead to disintegration of the wedge which has not happened.

Thus, while this paper offers a fascinating analysis under near-ideal conditions at field-experiment scale, it must be wondered if the glacier-ice support scenario can usefully be scaled up to the generality of RSF.
Here the debuttressing argument comes to the fore. From conversations with both Tim Davies and Sam McColl, the writer sees the logic of their reasoning - developed of course in respect of very large RSDs in alpine New Zealand troughs - that glaciers are plastic and thus also deformable, and that a failed slope can sag extensively even while in contact with the glacier, and remain in that metastable position after ice withdrawal. The writer has not refreshed and updated the course of this debate, but recalls that this 'debunking of debuttressing' may have been rowed back from to some degree.
Its relevance here, in considering scalability, is that it seems intuitively more reasonable for a large glacier to hold a small RSF in place, controlling its translation, than for a waning glacier to restrain and control a large RSF. If so, the less scalable are the findings of this paper, and the more likely it is that medium and larger RSFs have not been ice supported, and could thus be synchronous with or later than final deglaciation.

We make no claim that the ice-supported scenario can be scaled up. This is just the reviewer's own thought process, and we agree that many large RSFs were likely not ice

supported.  It is our belief that our manuscript is a useful contribution to the issue of the mechanics of RSF with and without the presence of ice.  We have demonstrated that one specific RSF had to have been ice supported.  We do not claim any scaling up to larger failures, nor across a geographic region.  It is clearly not going to be possible to make more sweeping generalizations until many more stability analyses have been published for a range of types of failure.  It is for this reason that we have refrained from making claims about our findings that cannot be sustained at this time.

**3 - the single date issue**

Recourse to the journal website and its record of 'submission-in-progress' (a novelty for this writer) indicates that the single-date issue has led to a previous rejection and has already been taken up by the Associate Editor as requiring greater circumspection. The Supplementary files address this issue, but add little to the simple fact that 'dates cost money' especially when much bulkier samples are required to obtain datable material than with the usual quartz-knob method (the writer has assisted CK Ballantyne in homing in on such quartz-knobs, and has also sampled Dartmoor tors for conveniently back-packable flake sizes - but RSFs are rarely in quartz-rich granite alas).

The writer has discussed with Derek Fabel the vexed question of number of samples required for reliable dating. He suggested, ideally, collecting ten, processing the first five, and if statistically consistent, calling a halt (if not, carry on). Given the cost, he accepted that four or even three might give a close-enough approximation to the actual age. Ballantyne has published a set of rock avalanche ages for the Highlands and Ireland based on three dates per site, with the protocol that if two are close and one an outlier, that can be rejected (IS Evans confirms that this is statistically invalid, as the single date could be 'more right' than the pair).

Here, the author admits that a single date is not really adequate to give a reliable age, and that the headscarp date may be little more than a limiting age. However the writer has further concerns, which would persist even with (say) three dates (and even once the precise locations of the two single samples were provided and inspected) :

We welcome the opportunity here to explain the issue of limited sampling.  It would not be appropriate to add this additional text to a manuscript.  We are fully aware of the requirements for reliable terrestrial cosmogenic dating.  Note that two of the authors are specialists within this field from a dedicated cosmogenic laboratory at the University of Aarhus.  The RSF lithology is Silurian muddy fine-grained sandstone.  It yields sufficient quartz to obtain a reliable date but only after a large lab effort.  The cleaning required to process just two samples was an excessive number of days and no grant giving body would consider the funding that would be required to cost the time involved in multiple sample preparation.  Given that we anticipated this problem we obtained two samples: one from the outer side (the riser) and one from the failure plane behind the RSF.  Thus, at a minimum we have a date for the first exposure of the riser to cosmic rays and similarly for the first exposure of the failure plane.  As reported within the manuscript, these two samples provided dates that are 'sensible' and interpretable within defined uncertainty ranges.  The sample taken from the riser is from an (ice?)-smoothed surface.  In principle it would have been possible to take more samples across the face of the RSF, but experience of similar sampling elsewhere indicates that applying the same procedures would result in

very similar closely aligned dates from several samples. The failure plane, as we note in the manuscript, is losing material due to post-glacial weathering and there are no other sites that we considered suitable for sampling. Sampling the headwall behind the RSF is not possible as it is heavily weathered and has been subject to post-glacial spalling, as the screes witness. Given that the date for the first exposure of the outer face of the RSF is consistent with multiple cosmogenic dates for erratic granite boulders nearby (as stated in the manuscript and reference provided), it seems sensible to place this date on the record by publication, with suitable caveats being given within the Supplementary Information. The exposure age for the failure plane is younger than expected if the plane had not been subject to spalling. However, as frequent spalling is evident it is reasonable that the date for the failure plane is younger than that of the riser and this too is placed on the record. Overall, despite the uncertainty related to having just two dates, the temporal consistency is a spur to using cosmogenic dating of the risers and failure planes of other RSFs to obtaining the timings of these events for which, overall, we have poor understanding of the time controls.

-- the source area is not a simple, visible rock scar of homogenous form or character. As discussed above, it is unclear whence the putative slipmass has come, but several sampling points across the 100-200 m width of bold crag - grassy bay (does it possess outcrops ?) - weaker crag would be desirable;

The issue of the source of the RSF has been addressed above.

-- the 'riser' is likewise a staircase of treads and risers, of varying boldness laterally and vertically, which should be sampled to reflect average conditions;

The riser is not a series of steps but in the vicinity of sampling it is a smoothed surface so multiple sampling is not helpful.

-- there has been considerable wastage along these mini-risers, as mentioned, and it may be difficult to judge what facet(s) truly represent the unmodified post-ice surface, thus several sampling points are needed.

An ice?-smoothed surface was sampled which has not been subject to significant post-ice modification.

Ideally, it would be intriguing to sample vertically down both source scar and riser, to see if they 'young' downwards with progressive slipmass displacement and ice surface lowering;

Given the cost involved in sample preparation for this specific lithology, a case for funds from any grant giving body would be difficult to make.

5 this was proposed by the writer for the Dartmoor tor dating project, where a tor had been quarried into, thus revealing a profile through the former ground surface and on to a depth below cosmo-penetration; the principle was accepted...
the thick scatter of coarse angular blocks on the riser could be dated, to give an earliest age for its separation from the rim to reveal the extant crags.

Due to post-glacial weathering of the headwall Holocene dates would be obtained which are not relevant to the study.

Possibly Schmidt hammer dating would give affordable insights into some of these concerns, as conducted for the Burtness Comb UD-LD by Peter Wilson with some success.

We have used a Schmidt hammer in other studies and in other contexts. We considered and dismissed Schmidt hammer relative dating due to the friability of the bedrock and the lack of a range of cosmogenic dates to anchor the Schmidt hammer readings.

 In this light, the two dates currently available deserve mention as broadly supportive, with all due caveats, and as advancing the case for more systematic dating of this site and others to compare or contrast.

Thank you for noting that our two dates should be put on record and publicly available within a publication. We have already included caveats to our two dates within the existing manuscript.

---

## Author Comment (AC2)

Review of Ms "Ice buttressing-controlled rock slope failure on a cirque headwall, English Lake District" by Carling et al.

Author responses to the reviewer's comments are given in blue text.

Ms # esurf-2023-14

**Introduction** This Ms proposes that a displaced but largely intact rock mass in Great Coum, Cumbria reached its present position by long-term, slow translation following failure, its motion being controlled by gradual debuttressing during post-LGM cirque glacier retreat. Two surface exposure dates, one (ca 18 ka) from the surface of the displaced mass and one (ca 12 ka) from the failure surface exposed by its motion, are claimed to constrain the timing of this event to ca 18 ka, which would be in line with established local deglaciation chronologies. The implication is that many of the undated intact RSF deposits in Britain are likely to have been retarded in their motion by gradual debuttressing by retreating ice.

We thank Prof. Tim Davies for this summary and the recommendation (at the end of the commentary) that we should submit a revised manuscript. However, we do not imply that the results of this specific investigation necessarily can be applied more widely as the reviewer avers. We do not claim that other UK RSFs were retarded by ice buttressing. Rather we present this study to illustrate the possibility that some RSFs were ice buttressed to spur further numerical investigation of RSFs.

This is a careful and detailed work that, irrespective of whether it is correct, would greatly benefit the community if published. The following comments suggest some further considerations that, in the opinion of the reviewer, would enhance the credibility of the work if addressed.

Thank you for noting that the work should be published to the benefit of the RSF community.

**Review** The critical evidence that supports the assumed gradual displacement of the rock mass is the fact that is substantially intact following a vertical displacement of ca 110 m on a slope of slightly greater than 30° (Fig. 4A, lines 644-5); however, there is stated to be an area of disintegrated rubble below the intact mass (lines 400-1) making up about 3% of the mass, so some degree of disintegration may well have occurred during emplacement.

If this slow displacement is accepted, then a cause can be provided in the form of downwasting of a cirque glacier assumed to have been present in Great Coum during the Last Glacial Maximum, so that the lateral support provided by the ice to the outer surface of the detached rock mass prevented it from displacing rapidly. This assumption is confirmed by detailed stability calculations for the in-situ rock mass, assumed to be detached from the parent mass by a planar failure surface in the form of a mapped fault; these show that the factor of safety F for the detached rock mass is in the region of 0.74-0.94 without ice support, whereas with ice support it is greater than 1.0.

The cosmogenic date of 18 ka for the outer surface of the displaced rock mass is asserted (lines 693-4) to be "compatible with the RSF movement during the final deglaciation around 19.2 to 16.6 ka". This is undoubtedly true; however, approximately the same date would have been found if the RSF had NOT moved at that time. There is

no doubt that the rock mass moved – *but there is no direct evidence that it moved at ca 18 ka*. However, if ice buttressing is proven to have been necessary to explain proven slow RSF motion, then uncovering of the RSF surface would have occurred during deglaciation of the cirque as hypothesised.

It is of course true that the RSF could have occurred later than the first exposure of the outer face to cosmic rays but, adopting Occam's Razor, given there would be no support for the failure after the ice burden was removed, then associating the failure with 18 ka seems reasonable. We also demonstrate that the RSF would be unsupported without ice and therefore descent would occur approximately at the same time as the outer face was exposed (see response to **Slow displacement of the RSF** below) Note we report the uncertainty of this date as ± 1.2 ka.  It is possible that the wedge of rock, that latterly constituted the RSF, was actually released from the headwall earlier than 18 ka and was supported by ice until the ice receded *c*., 18 ka.  However, this scenario is less likely as the thin wedge would probably have broken up within the ice body over time.

The hypothesis of the Ms therefore rests on (i) the assumption of slow displacement of the RSF, (ii) the interpretation of cosmogenic dating, and (iii) the stability analysis. These are now critically examined.

**Slow displacement of the RSF** In lines 404-406 Carling et al. indicate (correctly) that the H/L value for the event of 0.6 indicates no excessive runout, presumably to infer that this was a slow event; the largely intact nature of the RSF deposit leads to the same conclusion. But it is well-known that rapid blockslides can occur and remain intact at much lower values of H/L (e.g. Davies et al., 2006), meaning that *the fact that the deposit is intact is not a reliable indication that the movement was slow*. It is therefore quite possible that the RSF was emplaced rapidly (meaning at sliding speed commensurate with the marginal slope gradient – perhaps on the order of cm/sec to m/sec) and therefore it could have occurred after retreat of ice from its outer face – perhaps many thousands of years after.

The reviewer's statement is correct in principle.  It cannot be known *post hoc* if the RSF descended rapidly or slowly. This issue can only be resolved using physical evidence and numerical modelling.  We present two lines of physical evidence: 1) The short runout despite a steep failure plane, and 2) the remarkably undeformed fissile strata which are thinly bedded and which constitute a thin wedge. Undeformed strata might occur if the wedge had been very thick but that is not the case. Taken together we believe our case for slow descent is reasonable.

**Cosmogenic dating** As outlined in lines 614-6, the exposure date from the exposed headscarp (12 ka ± 0.8 ka) significantly postdates that from the outer face of the deposit (18.0 ± 1.2 ka). This requires that the upper headscarp became exposed about 6000 y after the disappearance of ice from the upper, outer face of the RSF. Carling et al. state (lines 614-620) that this is expected "… due the basal failure plane being progressively exposed after the upper portion of the RSF (where the sample OSF occurs) was clear of ice cover and the RSF began to move downslope" (note, however, that they later state (lines 695-7) "We interpret the much younger exposure age on the fault plane as the result of postglacial weathering and erosion.", suggesting some lack of confidence in this issue. The Supplementary Material indeed states that these factors have been considered in arriving at the quoted age). This means that the RSF had only displaced by about 20 – 30 m in the 6000 years following lowering of the cirque ice surface to expose the RSF outer surface. This requires extraordinarily slow downwasting of the

cirque ice – is it compatible with what is known about post-LGM and pre-YD deglaciation rates? Line 732 states that "…the Lake District was essentially ice-free by 14.7 ka…", and Great Coum is very low at 262-468 masl, so expecting its cirque glacier to have continued to support the RSF until later than 12 ka seems to be stretching the point too far, especially given that the Younger Dryas stadial onset was 12.9 ka (line 62). From this perspective the younger cosmogenic age is difficult to reconcile with the Ms hypothesis.

The reviewer presents careful reasoning here, but their interpretation is incorrect, possibly because we did not make our points sufficiently explicit.  We do not lack confidence in our interpretation, but we note in the manuscript that the fracturing and weathering on the failure plane would produce an exposure age on the failure plane likely younger than that of the outer face.  We have revised the manuscript to make our interpretation of the dating clearer.  Sample HW (from the failure plane, not the headwall) was 67 m above sample OSF (from the outer face of the RSF).  However, sample HW is only 48 m above the tread of the RSF.  Either way, the RSF has to descend only some 50 m or so to expose the failure plane which, given the uncertainty associated with the exposure age, would have been exposed at the same time as the RSF occurred, 18 ± 1.2ka.  As noted in the text, the failure plane is fractured and weathered whereas the outer surface of the RSF is planar and in places smoothed (possibly by ice).  We anticipate fracturing and weathering of the failure plane, hence a 2 m loss of material from the plane at the time of failure due to post-glacial processes is reasonable (see Supplementary Information). An exposure age much younger than 18 ka is to be expected, as post-glacial spalling of the bedrock surface is clearly an ongoing process at this site.

**Slope stability analysis** This appears to have been done with admirable attention to detail, and Carling et al. correctly emphasise the uncertainties involved (line 508). Nevertheless, in such work, particularly in the context of natural slopes where the subsurface composition is unknown and has to be inferred from surface exposures, it is necessary to include an error analysis to demonstrate the possible imprecision of the resulting factors of safety. For example, in section 5.3, F values are given to two decimal places with no indication of possible error. Further, in lines 518-9 reference is made to "marginal" F values of 1.07 to 1.22, suggesting that F = 1.22 is so close to 1.0 that it should be considered marginal. If this is the case (and it seems reasonable to me that there might be ~20% error in F values), then F = 0.78 should also be treated as marginal. In this perspective the factor of safety analysis becomes much less convincing in indicating that the RSF would have been unconditionally unstable in the absence of ice buttressing.

Thank you for noting the attention to the detail that we brought to this analysis.  We had anticipated that without this level of attention our results might be called into question. We did not include a tabulated error analysis for the three cases precisely because of the inherent uncertainty that exists with this kind of modelling. Rather, we varied parameters typically by ± 10 % (sometimes greater) to explore the potential uncertainty across thousands of simulations, which we argue serves as a reasonably robust uncertainty analysis.  We noted in the text that failure occurs in almost all cases unless ice buttressing is applied.  Note that reported F values (*e.g.*, 0.78) for unsupported scenarios are the largest values possible for reasonable wedge configurations, so these values are not marginal (*i.e.*, the uncertainty does not approach 1). We have edited the manuscript to make this point clear. It is well known that the value of F is only a guideline to slope stability and that a slope that is modelled as stable or otherwise might be

influenced by unknowable issues as well as by poor parameterization.  We argue that moving from scenarios without ice buttressing (whereby F is always < 1) to an ice-buttressed scenario (where F is always > 1) is indicative of a shift in rock mass precarity towards stability.

Other inherent uncertainties in the stability analysis include

- The assumption that the rear boundary of the RSF is a (wavy) planar fault with specified cohesion and friction coefficient, with no asperities or rock bridges. Particularly at first failure it is quite likely that this was not the case, meaning that the estimated F values for the non-buttressed case could be too low.

As noted in the text at lines 641 to 645 'The waviness number calculated from field data and applied in the model is low, which increases the propensity for failure. Preliminary trials showed that to stabilize unstable model slopes would require the use of unrealistically large waviness numbers (Miller, 1988) and so the waviness number was not varied in sensitivity analyses.'  For reasons of brevity, we did not expand our argument in the text.  Local rock bridges can increase the stability of a rock mass (see Yu et al., 2013 Frontiers in Earth Science, DOI 10.3389/feart.2023.1209259) but the issue becomes complex and beyond the scope of simple models that currently can be applied to RSFs.  Local enhanced bridging within the thinly-bedded thin wedge would have led to piecewise failure of the rock mass and progressive disintegration.

- Pore water pressure is treated rather offhandedly in the analysis, presumably because of lack of information. Assuming zero pwp would be the conservative assumption, given the stated fractured nature of the rock and the proximity of the failure surface to the exposed RSF face.

We argue that our consideration of pore water effects is not 'offhand'. We agree that it cannot be parameterized, so we considered the effects in those marginal cases whereby the stability of the rock mass might be influenced by pore water pressure and reported accordingly.

- It has been assumed that the ice surface of the cirque glacier is solid and horizontal throughout. This seems unlikely – my recollection of cirque ice is that it tends to slope very steeply up to a deep bergschrund where it contacts the rock. The presence of other crevasses is also likely to complicate the transmission of ice pressure to the RSF outer face.

We did not assume a horizontal ice mass and solidity is not an issue. Rather, we considered what ice loading force was required to stabilize the RSF and we present two cases.  We calculate the load as the minimum required to maintain slope stability and note that the load can be applied variably across the outer face of the RSF.  In this way we demonstrate that the amount of ice required to support this RSF is not large – this 'modest accomplishment' is all that we claim to demonstrate.  We mention already in the text the issue of a bergschrund being present and note that the multiplicity of ways the load might be applied.

Thus, while the factors of safety appear to conclusively support the Ms hypothesis at first sight, the lack of information on likely imprecisions means that the outcome is unconvincing.

We regret that the reviewer does not find the outcome convincing. Perhaps the revised MS is convincing.  Imprecision was dealt with by running thousands of simulations, the basic detail of which are summarized within the text, leading to most likely scenarios that trend from F<1 to F>1.  We acknowledge that with additional field data (impossible to obtain) a different outcome might be achieved.  However, given our field data and the model framework we believe that our interpretation of the values of F are defensible. We hope that our findings will stimulate further numerical modelling of RSFs to elucidate whether or not ice support occured in RSFs elsewhere.

**Summary** The above considerations cast some doubt on the certainty of the Ms hypothesis. For example, a plausible alternative hypothesis is that the RSF, having emerged from its ice cover at 18 ka, remained stable (with F > 1 but decreasing due to weathering) until it failed about 12 ka and translated while restrained only by basal and lateral friction, remaining largely intact as it did so.

We appreciate the reviewer's very careful consideration of our submission which we have found useful in preparing our revised MS. We agree that the scenario that the RSF might have occurred long after deglaciation must be considered, and we did consider that.  However, we find implausible the fact that the thin-bedded thin wedge of rock did not break up when translated down a steep failure plane.

**Recommendation**

Revise Ms to address these comments (or rebut them) and resubmit.

**Reference**

Davies, T.R.H., McSaveney, M.J. and Beetham, R. D. (2006). Rapid block glides – slide-surface fragmentation in New Zealand's Waikaremoana landslide. Quarterly Journal of Engineering Geology and Hydrogeology, 39: 115–129.

**Further comments** (refer to annotated Ms)

To be addressed in a revised submission

- Abstract lines 29-31: Stability analysis considers only the situation up to failure – it tells us nothing about whether post-failure motion was "catastrophic" or not

  Text edited to correct

- Line 310: omit "and"

  deleted

- Line 404: the word "rapid" states that all events with low H/L are rapid, which is probably true; however the implied converse, that all events with high H/L are slow, only applies when "slow" is means of the order of cm/minute to

cm/second. It does not imply "extremely slow" as is hypothesised for Great Coum

We agree with these statements, but the text referred to by the reviewer is only factual statements. No interpretation is made at this point that these ratios imply extremely slow decent of the RSF only that the runout is limited.  Latterly we argue that the small runout is possibly indicative of a slow descent.

• Line 508: yes there is parameter uncertainty – it needs to be quantified.

We addressed this issue at lines 289 to 394 and at lines 450 to 459 where the uncertainty is quantified in terms of ranges.  Altogether there were *c*., 30,000 model runs varying parameters, to explore the uncertainty. The main text has been revised to make it clear that the reported F values for the unsupported case are the maximum possible (see comments above for detail).

• Lines 556-8: Incorrect use of tonnes as unit of force

This is not the force applied by the ice, but the tonnage of ice required to supply the retaining force.

• Lines 600-605: Confusion in reference to "cirque erosion"

Text rewritten

• Lines 660-1: variation of ice strength with temperature is probably negligible in the context of the real precision of the present analysis

We agree with the reviewer's opinion.  However, we felt that the issue of ice temperature might be in the mind of some readers and felt some comment was necessary. Note that temperature is not a parameter within the modelling.

• Lines 705-707: Not all glacial ice carries erratic

We agree, but within the context of the Great Coum within the Lune Gorge the absence of glacial erratics is indicative of northern ice not entering the cirque (see Carling et al., 2013 cited in the manuscript for context).

---

## Author Response (AR1)

**Manuscript changes following reviewers' comments**

Lines 48 to 50: number of RSF updated and reference added (Referee 1).

Line 63: 'few' changed to 'one' (Referee 1).

Line 66: 'lower' deleted in accord with Referee 1.

Line 100: caption changed in accord with suggestion to label the ice flow phases (Ice flow phases also labelled on Fig. 2) in accord with Referee 1.

Line 166: 'sandstones' added in accord with the suggestion of Referee 1.

Fig. 4: the word 'debris' has been made more clear in accord with Referee 1.

Fig. 4B: the notation related to the cosmo samples has been added.

Line 299: text has been added to emphasize that the RSF sampling site was likely ice-smoothed.  A supplementary figure (Fig. S3) has also been added to show this clearly, in accord with Referee 1.

Line 522-523: text has been added to make it clear that factors of safety higher than stated cannot pertain to this analysis.  This is in accord with Referee 2 who queried the uncertainty of reported F values.

Line 526: text added to indicate that meltwater effects the water pressure in accord with query from Referee 2.

Caption to Fig. 7 has been edited to make it clear that the inner edge of the pentahedral is irregular to depict potential irregularity in the ice loading at the suggestion of Reviewer 1.

Lines 609 and 614: Minor changes to the text have been made due to query from Referee 2 about timing of active erosion in the cirque.

Line 631: text added to make it clear that the cosmogenic sampling location is on the failure plane of the RSF. In response to queries from both Referee 1 and 2.

Line 653: the word 'rock' has been added to define the obstacle to downward motion of the RSF, as a result of a query from Referee 1 with regard to the nature of any obstacle.

Lines 703 and 705 to 707: Text changed to emphasize which cosmo sample we are reporting and the fact that the failure plane is fractured whereas the outer face is not.  In response to both queries from Referee 1 and 2.

Line 1146: Reference added at the suggestion of Referee 1.